# How heating tracers drive self-lofting long-lived stratospheric anticyclones: simple dynamical models

Kasturi S. Shah[1,2] and Peter H. Haynes[1]

[1]Department of Applied Mathematics and Theoretical Physics, University of Cambridge, Cambridge CB3 0WA, UK
[2]Department of Earth, Atmospheric and Planetary Sciences, Massachusetts Institute of Technology, Cambridge MA 02139, USA

**Correspondence:** Kasturi Shah (kss58@cam.ac.uk)

**Abstract.**

Long-lived 'bubbles' of wildfire smoke or volcanic aerosol have recently been observed in the stratosphere, co-located with ozone, carbon monoxide, and water vapour anomalies. These bubbles often survive for several weeks, during which time they ascend through vertical distances of 15km or more. Meteorological analysis data suggests that this aerosol is contained within strong, persistent anticyclonic vortices. Absorption of solar radiation by the aerosol is hypothesised to drive the ascent of the bubbles, but the dynamics of how this heating gives rise to a single-sign anticyclonic vorticity anomaly has thus far been unclear. We present a description of heating-driven stratospheric vortices, based on an axisymmetric balanced model. The simplest version of this model includes a specified localised heating moving upwards at fixed velocity and produces a steadily translating solution with a single-signed anticyclonic vortex co-located with the heating, with corresponding temperature anomalies forming a vertical dipole, matching observations. A more complex version includes the two-way interaction between a heating tracer, representing the aerosol, and the dynamics. An evolving tracer provides heating which drives a secondary circulation and this in turn transports the tracer. Through this two-way interaction an initial distribution of tracer drives a circulation and forms a self-lofting tracer-filled anticyclonic vortex. Scaling arguments show that upward velocity is proportional to heating magnitude, but the magnitude of peak quasigeostrophic vorticity is $O(f)$ ($f$ is the Coriolis parameter) and independent of the heating magnitude. Estimates of vorticity from observations match our theoretical predictions. We discuss 3D effects such as vortex stripping and dispersion of tracer outside the vortex by the large-scale flow which cannot be captured explicitly by the axisymmetric model and are likely to be important in the real atmosphere. The large $O(f)$ vorticity of the fully developed anticyclones explains their observed persistence and their effective confinement of tracers. To further investigate the early stages of formation of tracer-filled vortices, we consider an idealised configuration of a homogeneous tracer layer. A linearised calculation reveals that the upper part of the layer is destabilised due to the decrease in tracer concentrations with height there, which sets up a self-reinforcing effect where upward lofting of tracer results in stronger heating and hence stronger lofting. Small amplitude disturbances form isolated tracer plumes that ascend out of the initial layer, indicative of a self-organisation of the flow. The relevance of these idealised models to formation and persistence of tracer-filled vortices in the real atmosphere is discussed and it is suggested that a key factor in their formation is the time taken to reach the fully-developed stage, which is shorter for strong heating rates.

# 1   Introduction

Smoke from wildfires in Australia in 2020 has been observed to enter the stratosphere and, unexpectedly, to form long-lived coherent anomalies that persist for several weeks and that ascend through distances of several kilometers (Khaykin et al., 2020; Kablick III et al., 2020). The smoke anomalies are co-located with anomalies in chemical species such as ozone, carbon monoxide, and water vapour (Kablick III et al., 2020; Khaykin et al., 2020). Furthermore, meteorological analysis fields constructed from a combination of in-situ and satellite data, processed in conjuction with weather prediction models, have suggested the smoke anomalies are co-located with strong anticyclonic vortices (Khaykin et al., 2020; Kablick III et al., 2020). Similar long-lived ascending smoke-filled vortices have been identified following the Canadian wildfires in 2017 (Allen et al., 2020; Lestrelin et al., 2021) and corresponding aerosol-filled vortices have been identified following the Raikoke eruption in 2019 (Khaykin et al., 2022) which penetrated the stratosphere. Doglioni et al. (2022) have demonstrated, for the 2017 Canadian wildfire case, the successful simulation of smoke-filled vortices in a chemistry-climate model that includes the heating effects of the injected smoke.

This apparent co-location of tracers with coherent vortices is consistent with a physical interpretation where strong vortices effectively act to isolate tracers (which may be chemical species, small particles such as wildfire smoke or volcanic aerosol [henceforth collectively referred to as 'aerosol'], etc) from their environment, thus maintaining large anomalous concentrations that would otherwise be reduced through the effects of mixing. The idea of vortex isolation in the stratosphere has previously been much discussed in the context of the winter polar vortex, for example, the isolation of regions of low ozone concentration in the austral spring lower stratosphere following chemical destruction (e.g. McIntyre, 1989), or regions of low concentration of Pinatubo aerosol in the polar vortex in boreal winter 1992 (Plumb et al., 1994). Vortex isolation has also been discussed in the context of smaller scale vortices such as the 'frozen-in anticyclones', which have been observed in the mid- and high-latitude summer stratosphere and originate at low latitudes (Manney et al., 2006; Allen et al., 2011).

Much of our physical understanding of vortex isolation originates in the studies of two-dimensional turbulence, where it has been observed that flow self-organises into strong, relatively long-lived coherent vortices of different signs (Boffetta and Ecke, 2012). The vorticity anomalies associated with the vortices themselves are quasi-circular; whereas in the region outside of the vortices, the vorticity field is filamental and relatively passive (since the flow associated with small-scale vorticity anomalies is weak). The evolution of the flow is therefore largely governed by interactions between the coherent vortices. A passive tracer concentration field has a similiar character to that of the vorticity field. Within the vortices, anomalies in passive tracer concentration remain relatively large. Outside the vortices the concentration field is stretched into filaments, promoting mixing and hence decay of concentration anomalies. These mechanisms identified in two-dimensional turbulence can be extended

in ways that are relevant to vortex evolution in the stratosphere. Three-dimensional flows that are close to hydrostatic and geostrophic balance can be described by the quasigeostrophic potential vorticity (QGPV) equation, which is very similar to the two-dimensional vorticity equation, with the QGPV rather than the vorticity being advected by the horizontal flow, except that the QGPV anomalies vary in the vertical and the horizontal flow on each level is determined by the QGPV field over a range of levels. The effects of a structured large-scale flow, as exists in the stratosphere, can be incorporated by considering vorticity anomalies in 2-D or QGPV anomalies in 3-D subject to an externally imposed shear or strain field. It may be shown in both 2-D (e.g. Kida, 1981) and, with the effect of vertical shear included, QG cases (Meacham et al., 1994) that when the external shear or strain is sufficiently weak then the vortex remains coherent. However, when the external shear or strain is strong then the vortex is strongly deformed and no longer remains coherent. Since vorticity, or potential vorticity as its generalisation for rotating stratified flow, is itself a tracer, the same principles will apply to any tracer anomaly contained within the vortex. For as long as a vortex survives, it is likely to be an effective isolator of tracer anomalies. If a vortex is destroyed by strong deformation then any tracer anomalies starting within the vortex will inevitably, along with vorticity anomalies, be stretched into filamentary structures and dissipated (in the case of the tracer, by mixing processes).

Mariotti et al. (1994) have further identified the phenomenon of 'vortex stripping' where, when a 2-D vortex containing a continuous range of vorticity values is subject to modest horizontal external deformation, outer layers with smaller vorticity magnitudes are stripped away but the interior, where the vorticity magnitude is larger, may persist and remain coherent. The inevitable consequence for the situation where a passive tracer anomaly is originally co-located with the vortex will be that outer layers of tracer are stripped away but the tracer in the interior will persist. While there do not seem to have been any specific extensions of the Mariotti et al. (1994) vortex stripping experiments from two-dimensions to the quasigeostrophic case, the analogy between the evolution found by Kida (1981) in 2-D vortex dynamics and that found by Meacham et al. (1994) in QG dynamics, makes it seem very likely that a similar phenomenology will apply in QG, with vertical shear as well as horizontal deformation playing a role.

Therefore, in one sense, the existence of a persistent aerosol (i.e. tracer) anomaly within coherent vortices, as recently observed, is expected from the above physical description. However much of the previous discussion of persistent coherent vortices, both in the atmosphere and the ocean, has emphasised the material conservation of potential vorticity (PV) and focused on whether the vortices can survive external perturbation. In the case of the recently observed smoke-containing stratospheric vortices, it has been noted that *non*-material-conservation of potential vorticity is a key ingredient. The reason is that the observed ascent of aerosol anomalies and the vortices themselves imply a substantial change in potential temperature of the fluid containing those anomalies, requiring substantial diabatic effects arising from radiative heating due to sunlight

absorption by smoke particles (Sellitto et al., 2023) or volcanic aerosol (Khaykin et al., 2022). Under these circumstances, potential vorticity is not materially conserved. Furthermore in the presence of diabatic effects it cannot be considered that potential vorticity is simply transported in the vertical across isentropic surfaces in the same way as a passive tracer (Haynes and McIntyre, 1987). As noted by Kablick III et al. (2020), this prevents an interpretation of the vortices as bubbles of air that originate in the troposphere, with low (absolute) values of PV, along with high concentrations of smoke particles and low concentrations of e.g., ozone, and that simply preserve their low values of PV, along with their smoke and ozone concentration, as they ascend through the stratosphere. The dynamics of these coherent, long-lived vortices and, in particular, the precise role of radiative heating due to aerosol in their formation and evolution remains uncertain and requires further examination. A promising approach to describing the dynamics is to consider the evolution of the PV field, and then, under the assumption that the flow is balanced, use PV inversion to determine other dynamical variables, such as temperature and velocity (Hoskins et al., 1985).

The previously cited papers on aerosol-filled vortices (Lestrelin et al., 2021; Khaykin et al., 2020; Kablick III et al., 2020; Khaykin et al., 2022) have presented descriptions of the PV structure based on meteorological analysis or reanalysis data, whilst emphasising that this is subject to significant uncertainty. For example, the ERA5 reanalysis (Hersbach et al., 2020) does not incorporate aerosol emissions from wildfires and volcanoes, nor satellite measurements of aerosol extinction. The PV structure there arises from the limited information available about the temperature distribution: ERA5 is only informed about anomalous heating by the assimilation of IASI infrared radiances and GPS radio-occultations which constrain the temperature (see Khaykin et al., 2020; Lestrelin et al., 2021, who discuss this in more detail), and furthermore the PV distribution is relatively poorly constrained by the temperature distribution. However within the limitations of the estimation of PV, all the above studies identify the structure of the vortices as a *single*-sign anticyclonic PV anomaly whose centroid is roughly co-located with the centroid of the aerosol anomaly. There is some supporting evidence for such a structure from the Doglioni et al. (2022) modelling study, which is not subject to the same limitations as the reanalysis data.

This single-signed PV anomaly is quite different to the balanced dynamical response to specified localised (stationary) heating expected from simple theory for axisymmetric flow, which has been applied, for example, to extratropical anticyclones and cyclones in the troposphere and lower stratosphere (Thorpe, 1985) or tropical cyclones (Schubert and Hack, 1983). Such theory is equivalent in many ways (apart from the fact that the spatial scales are larger) to the zonally symmetric problem describing the dynamics of the zonally averaged circulation, such as the Brewer-Dobson circulation or the meridional circulation during dynamical events such as sudden stratospheric warmings (Dunkerton et al., 1981; Plumb, 1982). Determining the flow response to an axisymmetric applied heating or applied force is often called the 'Eliassen problem'. In a PV-based description,

a heating applied in a localised region provides a dipolar PV forcing, anticyclonic above the region of heating and cyclonic below. Therefore, if injection of aerosol into the stratosphere leads to a localised heating, the short-term effect is expected to be a pair of PV anomalies, anticyclonic above the heating and cyclonic below the heating, rather than a single anticyclone at the same level as the heating as is apparently observed.

Accordingly, key specific questions which motivate further study of the dynamics of smoke- or aerosol-filled vortices are: (i) How does an isolated anticyclonic vortex emerge as a response to heating and why is the anticyclonic vortex apparently centred at the same level as the heating rather than above it? (ii) What determines the rate of rise of the tracer anomaly and accompanying anticyclonic vortex? (iii) What determines the strength of the vortex and the corresponding temperature anomaly? (iv) Once aerosol is injected into the stratosphere, what is the mechanism for its organisation into long-lived ascending heating-driven vortex structures and under what conditions is this organisation likely to take place?

A non-standard aspect of the dynamics of the observed stratospheric vortices is that the heating field is determined by the aerosol which co-evolves with the dynamical fields. A simple representation is that the aerosol is a tracer field transported by the flow, and the heating is simply proportional to the tracer concentration. Solution of the Eliassen problem for a localised heating shows that, alongside the response in PV, there is also a secondary circulation response, which is upward motion in the centre of the heating region (Hoskins et al., 2003). Therefore if the heating is resulting from a localised tracer anomaly then the tracer, and correspondingly the heating, is expected to move upwards. For brevity, rather than using the terms 'smoke' or 'aerosol' we will use the term 'tracer', with it being understood that this tracer gives rise to a heating effect (and is therefore not 'passive' since it affects the dynamics). §2 of this paper sets out a simple dynamical model including these ingredients: dynamics driven by applied heating as described by the axisymmetric Eliassen problem, with the heating proportional to a tracer concentration that evolves with the dynamical fields. A simple addition is to include thermal damping, representing long-wave radiative transfer, that is determined by the temperature field. The full dynamical formulation can be simplified by making the quasigeostrophic (QG) approximation and it is this QG version of the dynamics that is mostly used in the remainder of the paper. The implications of non-QG dynamics are considered first in §2.1, but limited to the early-time evolution. As we submitted this manuscript, we became aware that an independent paper on the dynamics of heating-driven vortices, (Podglajen et al., 2024, henceforth P2024), also using a model including a heating tracer, had been submitted for publication elsewhere. Their paper avoids reliance on the quasigeostrophic approximation by combining a different theoretical approach with numerical calculations in a 3-D nonhydrostatic model. In revising this paper, we have incorporated various references to and comments on Podglajen et al. (2024).

In §3, explicit numerical solutions are presented for the evolution from initial conditions of a distribution of smoke that is localised in the horizontal and vertical. The highly simplified case of specified ascent of the smoke (and hence the heating) is considered first, followed by the fully interactive case, where the aerosol drives a secondary circulation through its heating effect and is transported by that circulation. Given the inability of the axisymmetric model to explicitly represent deformation by the large-scale flow and the consequent vortex stripping, a simple adjustment of the smoke to represent this effect within the axisymmetric formulation is also presented and discussed. Further aspects explored include the effects of thermal damping and non-Boussinesq effects. A summary of our key findings is provided at the end of §3.5, and a perspective on our study and Podglajen et al. (2024) in §3.6.

In §4, a different problem is considered in which the smoke is initially confined to a horizontally homogeneous layer. The geometry is assumed to be two-dimensional and periodic in the horizontal rather than axisymmetric. A linear stability problem is solved to demonstrate that this configuration is unstable as a result of the coupling between the smoke and the dynamics. Numerical solutions, under the QG approximation, can follow the evolution out of the linear regime and show how this coupling leads to self-organisation of the flow to give a discrete set of rising smoke plumes. §5 summarises the results and discusses the implications for the formation and evolution of smoke-driven vortices in the real atmosphere. It is argued that, whilst the axisymmetric or two-dimensional models have fundamental limitations, the conclusions obtained from these models can be combined with knowledge of two-dimensional or QG vortex dynamics from much previous work to give useful insights into the behaviour of the aerosol-driven vortices in the real 3-D atmosphere.

## 2 Dynamical model formulation

To describe the dynamics resulting from an aerosol-like tracer that generates diabatic heating and consequently anomalies in vorticity, temperature, and velocity, we consider an axisymmetric framework on an $f$-plane. The axisymmetric framework immediately neglects important ingredients mentioned above, including the 'vortex-stripping' effect of large-scale shear and strain fields; though in §3.4, we will consider the effect of a simple ad hoc representation of such effects in the axisymmetric model. The radial momentum equation is approximated by gradient wind balance and the vertical momentum equation by hydrostatic balance. We define $r$ a radial coordinate and $z$ a vertical log-pressure coordinate. The resulting governing equations

are,

$$\frac{\partial v}{\partial t} + u\left(f + \frac{1}{r}\frac{\partial(rv)}{\partial r}\right) + w\frac{\partial v}{\partial z} = G, \tag{1a}$$

$$-\left(f + \frac{v}{r}\right)v = -\frac{\partial \Phi}{\partial r}, \tag{1b}$$

$$\frac{1}{r}\frac{\partial}{\partial r}(ru) + \frac{1}{\rho_0}\frac{\partial}{\partial z}(\rho_0 w) = 0, \tag{1c}$$

$$\frac{\partial \Phi}{\partial z} = \frac{RT}{H_0}, \tag{1d}$$

$$\frac{\partial T}{\partial t} + u\frac{\partial T}{\partial r} + w\left(\frac{\partial T}{\partial z} + \frac{\kappa T}{H_0}\right) + w\left(\frac{\partial T_B}{\partial z} + \frac{\kappa T_B}{H_0}\right) = Q, \tag{1e}$$

representing the azimuthal momentum equation, gradient wind balance, continuity equation, hydrostatic balance, and the thermodynamic equation respectively. The notation is largely standard, with $(u, v, w)$ velocity components in radial, azimuthal and vertical directions, i.e. $(u, w)$ describes the secondary circulation in the $(r, z)$ plane. $T$ and $\Phi$ are horizontally varying parts of temperature and geopotential fields, and $T_B(z)$ is a background vertically-varying temperature field. $G$ and $Q$ are azimuthal

force and heating respectively, to be specified. Since the effect of an azimuthal force is not very relevant to this problem we shall neglect $G$ from now on. The density $\rho_0$ is equal to $e^{-z/H}$, $\kappa = R/c_p = 2/7$, and $f$ is the Coriolis frequency. A non-standard feature is the introduction of $H_0$ which may be different from $H$. Here, $H_0$ controls the stratification while $H$ controls the variation of density $\rho_0$. This allows the possibility of considering a 'Boussinesq-stratified' case with $\rho_0$ constant but retaining stratification, by taking $H_0$ finite but choosing $H$ to be very large.

As is standard, we use (1c) to define a mass streamfunction $\Psi$ for the secondary circulation such that $u = (1/r\rho_0)\partial\Psi/\partial z$ and $w = -(1/r\rho_0)\partial\Psi/\partial r$. This reduces the number of independent fields to three ($v$, $T$ and $\Psi$).

It is convenient to combine (1b) and (1d) to give the relevant form of the thermal wind equation

$$\frac{\partial}{\partial z}\left(\left(fv + \frac{v}{r}\right)v\right) = \frac{R}{H_0}\frac{\partial T}{\partial r}. \tag{1f}$$

The principles underlying the behaviour allowed by these equations are well-known. A first important implication is that

the flow in the $(r, z)$ plane may be determined instantaneously by eliminating $\partial_t v$ and $\partial_t T$ from (1a) and (1e) using (1f). This leads to a second-order PDE in $(r, z)$ for the mass streamfunction $\Psi$, with coefficients depending on $v$ and $T$ and a forcing term which is a combination of $G$ and $Q$. This PDE, the Sawyer-Eliassen equation, is elliptic and the problem of determining $\Psi$ is therefore well-posed, if the PV is everywhere non-zero with the same sign as $f$. In physical terms, the ellipticity requirement is equivalent to the requirement that the instantaneous azimuthal flow and accompanying temperature structure are inertially

stable. There is then sufficient information in the equations to evaluate $\partial_t v$ and $\partial_t T$, i.e. to advance the two fields $v$ and $T$ in time. However, because of the constraint (1f) there is actually only one field to be advanced in time with the other following

from thermal wind. (A technical detail is that if $v$ is advanced, $T$ must also be advanced in a limited way, e.g. at one level, and if $T$ is advanced, $v$ must correspondingly be advanced in a limited way.) Alternatively, a time evolution equation for PV may be derived from (1a) and (1e) which describes advection of PV by the flow in the $(r, z)$ plane together with forcing of PV due to the $G$ and $Q$ terms. Then a nonlinear PDE in $(r, z)$, expressing the principle of PV invertibility (Hoskins et al., 1985), can be solved to determined $v$ and $T$ in terms of the PV.

The condition for the PDE for $\Psi$ to be elliptic depends on the instantaneous $v$ and $T$ fields. Therefore even if the ellipticity condition is satisfied initially it may not remain satisfied. Indeed there are many examples in previous studies, for example in the tropical cyclone literature, where the breakdown of ellipticity limits the time over which the equations can be integrated and various ad hoc adjustments have been devised to overcome this (Möller and Shapiro, 2002).

A significant simplification is to make the quasigeostrophic approximation, which requires small Rossby number $Ro$, i.e. $|v| \ll fL$, where $L$ is a typical horizontal length scale, or equivalently $|\zeta| \ll f$, where the relative vorticity $\zeta = r^{-1}\partial_r(rv)$. The PDE for $\Psi$ is then linear and elliptic, and hence existence of solutions is guaranteed for all time.

To the dynamical equations presented above we add the evolution equation for a heating tracer, with concentration $\chi$. In the axisymmetric case, the tracer advection equation is

$$\frac{\partial \chi}{\partial t} + u\frac{\partial \chi}{\partial r} + w\frac{\partial \chi}{\partial z} = \frac{1}{r}\frac{\partial}{\partial r}\left(\kappa_h r\frac{\partial \chi}{\partial r}\right) + \frac{1}{\rho_0}\frac{\partial}{\partial z}\left(\kappa_v \rho_0\frac{\partial \chi}{\partial z}\right), \tag{2}$$

where $\kappa_h$ and $\kappa_v$ are respectively horizontal and vertical diffusivities (which may be functions of $r$ and $z$ if required).

The dynamical equations and the aerosol tracer equation are coupled by allowing the heating $Q$ to depend on the aerosol concentration $\chi$. It is convenient to separate $Q$ into two parts, one, $Q_s$, determined by the aerosol concentration, and the other, $Q_l$ to represent long-wave radiation, determined by the temperature anomaly. The latter is approximated in this study by a Newtonian cooling term, $-\alpha T$, where $\alpha$ is assumed constant, such that,

$$Q = Q_s - \alpha T. \tag{3}$$

Here, $Q_s$ represents heating due to sunlight absorption by aerosol and is therefore approximated as proportional to $\chi$. Indeed it is convenient to simply write $Q_s = \chi$, so that $\chi$ is, in effect, defined in units equivalent to implied heating rate. In some of the calculations to be presented below, $Q_s$ will simply be specified as a function of $r$, $z$ and $t$, to illustrate some of the physical mechanisms that operate.

The rest of this section will now focus on the equations resulting from the quasigeostrophic approximation. We briefly discuss the practicalities of solving the non-quasigeostrophic balanced vortex formulation in §2.1. The quasigeostrophic form

of the above dynamical equations (1a)-(1e) is

$$\frac{\partial v}{\partial t} + fu = 0 \tag{4a}$$

$$-fv = -\frac{\partial \Phi}{\partial r} \tag{4b}$$

$$\frac{1}{r}\frac{\partial}{\partial r}(ru) + \frac{1}{\rho_0}\frac{\partial}{\partial z}(\rho_0 w) = 0 \tag{4c}$$

$$\frac{\partial \Phi}{\partial z} = \frac{RT}{H_0} \tag{4d}$$

$$\frac{\partial T}{\partial t} + w\left(\frac{dT_B}{dz} + \frac{\kappa T_B}{H_0}\right) = Q = Q_s - \alpha T. \tag{4e}$$

There are two main differences to the full Eliassen model: the first is neglecting gradients in azimuthal wind in the quasi-geostrophic azimuthal momentum equation (4a), and the second is the reduced gradient wind balance (4b). The corresponding form of the thermal wind equation (1f) is

$$f\frac{\partial v}{\partial z} = \frac{R}{H_0}\frac{\partial T}{\partial r}. \tag{4f}$$

It is convenient to write $v$ and $T$ in terms of a quasigeostrophic streamfunction $\psi(r, z, t)$, such that $v = \partial_r\psi$, $T = (H_0 f/R)\partial_z\psi$, and then, from these equations, to derive a quasigeostrophic PV equation

$$\frac{\partial}{\partial t}\left[\frac{1}{r}\frac{\partial}{\partial r}\left(r\frac{\partial \psi}{\partial r}\right) + \frac{1}{\rho_0}\frac{\partial}{\partial z}\left(\frac{f^2}{N^2}\rho_0\frac{\partial \psi}{\partial z}\right)\right] = \frac{f}{\rho_0}\frac{\partial}{\partial z}\left(\frac{\rho_0 RQ_s}{H_0 N^2}\right) - \frac{\alpha}{\rho_0}\frac{\partial}{\partial z}\left(\frac{\rho_0 f^2}{N^2}\frac{\partial \psi}{\partial z}\right), \tag{5}$$

where the quantity in the square brackets on the left-hand size is the quasigeostrophic PV, $q$. The buoyancy frequency, $N$, is defined by

$$N^2 = \frac{R}{H_0}\left(\frac{dT_B}{dz} + \frac{\kappa T_B}{H_0}\right). \tag{6}$$

There is a corresponding equation for the velocities $(u, w)$, conveniently expressed as an equation for $w$,

$$N^2\frac{1}{r}\frac{\partial}{\partial r}\left(r\frac{\partial w}{\partial r}\right) + f^2\frac{\partial}{\partial z}\left(\frac{1}{\rho_0}\frac{\partial(\rho_0 w)}{\partial z}\right) = \frac{R}{rH_0}\frac{\partial}{\partial r}\left(r\frac{\partial Q}{\partial r}\right) \equiv \frac{R}{rH_0}\frac{\partial}{\partial r}\left(r\frac{\partial Q_s}{\partial r}\right) - \frac{\alpha f}{r}\frac{\partial}{\partial r}\left(r\frac{\partial^2 \psi}{\partial r \partial z}\right) \tag{7}$$

with $u$ to be deduced via (4c).

The equations (2), (5) and (7), together with a specification of $Q_s$ in terms of tracer concentration $\chi$ and suitable boundary conditions, form a complete model system to study the axisymmetric dynamics driven by heating due to an aerosol-like tracer. The quasigeostrophic approximation has been made to arrive at these equations, but an equivalent set under more general balanced dynamics would be obtained using the full equations (1a)-(1e) for the Eliassen problem. A brief discussion of the non-QG balanced vortex model is in §2.1, and solutions of the quasigeostrophic governing equations are presented in §3.

In the simulations to be reported below, we solve initial value problems starting from a configuration in which the flow is at rest (i.e. there is no initial PV anomaly) and the initial distribution of the tracer $\chi$ is chosen for simplicity to be Gaussian

$$\chi \propto Q_s = \chi_0 \exp\left(-\frac{1}{2}\left(\frac{r}{r_0}\right)^2 - \frac{1}{2}\left(\frac{(z - z_c)}{z_0}\right)^2\right). \tag{8}$$

This profile is plotted in red shading in Figure 3(a) for $\chi_0 = 5 \times 10^{-5}\mathrm{Ks}^{-1}$, $r_c = 0\mathrm{m}$, $z_c = 1 \times 10^4\mathrm{m}$, $r_0 = 2 \times 10^5\mathrm{m}$, $z_0 = 1.5 \times 10^3\mathrm{m}$. $r_0$ and $z_0$ have been chosen to be roughly consistent with the smoke-filled vortices described in Khaykin et al. (2020): for this functional form, $L$ is roughly 200km and $D$ is roughly 5km. The relevant Eliassen equations, either the non-quasigeostrophic form (1) or the quasigeostrophic form (4) are then integrated forward in time with $\chi$, and hence $Q_s$, as specified above. Podglajen et al. (2024) choose a different initial condition, with an initial distribution of tracer together with a non-zero low PV anomaly; we will return to a discussion of the initial condition in §5.

Parameter choices are as follows and will be retained in further sections unless otherwise stated. The Coriolis frequency is calculated at $45^o\mathrm{N}$, i.e. $f = 10^{-4}\mathrm{s}^{-1}$, $R = 287\mathrm{m}^2\mathrm{s}^{-2}\mathrm{K}^{-1}$, $H_0 = 7000\mathrm{m}$ and $H = 10^8\mathrm{m}$ so that it is effectively constant with height and the system is Boussinesq. (Vertically varying density is explored in §3.5.) The background temperature $T_B = gH_0/R$, and hence the buoyancy frequency $N$ are assumed constant, with $N^2 = g\kappa/H_0$ and hence $N \simeq 2 \times 10^{-2}\mathrm{s}^{-1}$ for the parameter values chosen. Horizontal and vertical diffusivities are $\kappa_h = 10^3\mathrm{m}^2\mathrm{s}^{-1}$ and $\kappa_v = 2\times10^{-1}\mathrm{m}^2\mathrm{s}^{-1}$ respectively.

## 2.1 Non-QG Eliassen balanced vortex evolution

On solving the non-quasigeostrophic balanced model equations, it is generally found that there is catastrophic breakdown of the numerical solution at some time and simple approaches such as reducing timestep do not resolve this problem. Overall, the early time evolution of the tracer (and hence the heating rate) in the quasigeostrophic equations matches the evolution of the non-quasigeostrophic equations: the tracer profile right before the non-QG numerical solution breaks down is plotted in Figure 1(a) for the non-QG and QG models. Based on the evolution of minimum PV in panel (b), it is the dynamical response to the heating that is different across the two models.

Examination of the minimum potential vorticity as this 'breakdown time' is approached shows the PV approaching zero (roughly as a linear function of time), suggesting that non-ellipticity is the cause of the breakdown (Figure 1(b)). Adding thermal damping lengthens the breakdown time, but, again, does not prevent the breakdown from occurring. Podglajen et al. (2024) provide a theoretical description of this breakdown which we discuss further in §3.6.

Loss of ellipticity is a familiar problem when solving the non-quasigeostrophic axisymmetric (or two-dimensional) vortex models, e.g. in studies of tropical cyclones. The physical interpretation is often that the axisymmetric flow becomes unstable to symmetric instability or inertial instability (Möller and Shapiro, 2002; Wirth and Dunkerton, 2006, 2009; Bui et al., 2009).

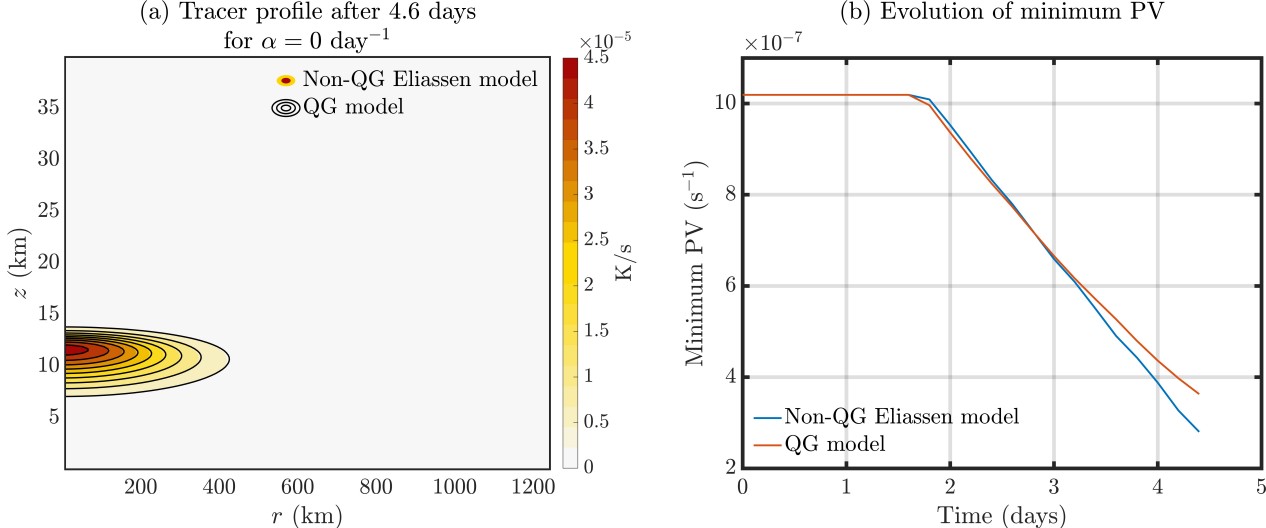

**Figure 1.** Solutions of the Eliassen problem, equations (1), and quasigeostrophic problem, equations (4), with no thermal relaxation. (a) Tracer cross-section after 6.9 days from the Eliassen balanced vortex model (colour shading) and quasigeostrophic problem (black contours), (b) minimum vorticity in balanced vortex model over time.

In order to extend the time for which models can be integrated such studies often implement ad hoc regularisation methods that can be justified as representing the adjustment under the effects of instability. One approach is simply to increase vorticity such that the potential vorticity remains greater than a specified threshold value and hence positive (Möller and Shapiro, 2002). Other studies use a relaxation towards an inertially neutral state to maintain small amplitudes of symmetric instability that prevent breakdown of the elliptic solver (Wirth and Dunkerton, 2006).

Given the ad hoc nature of these adjustment procedures, and some arguments in the tropical cyclone literature that different adjustments can give quite different outcomes for the evolution (Wang and Smith, 2019), we have chosen not to implement any such adjustments in this paper, and focus §3 on calculations using the quasigeostrophic equations (4).

## 3 Dynamical behaviour in axisymmetric framework

This section explores solutions of the axisymmetric quasigeostrophic formulation starting from the Gaussian initial condition (8).

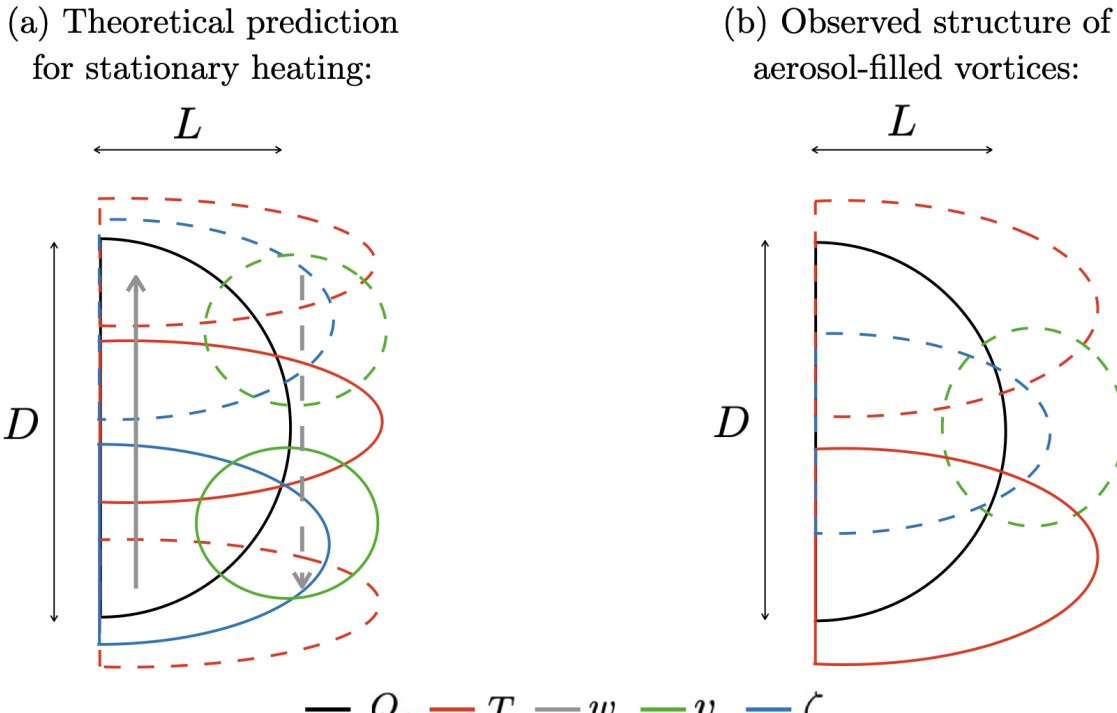

(a) Theoretical prediction for stationary heating:

(b) Observed structure of aerosol-filled vortices:

$$- Q_s \quad - T \quad - w \quad - v \quad - \zeta$$

**Figure 2.** Schematic diagram of (a) the theoretical prediction to specified localised stationary heating and (b) the observed structure of stratospheric vortices generated from wildfires (cf. Khaykin et al., 2020; Kablick III et al., 2020; Lestrelin et al., 2021) and volcanic eruptions (Khaykin et al., 2022). The heating tracer (black), temperature (red), vertical velocity (grey arrows), vorticity (blue), and azimuthal wind (green) are drawn, where solid contours represent positive values and dashed contours represent negative values. No arrows have been included in (b) because upward motion has not been directly observed. However the upward motion of the aerosol strongly suggests that the vertical velocity at the centre of the structure is positive. Also marked are the radial and vertical length scales, $L$ and $D$ respectively, of the specified localised heating.

## 3.1 Predicted versus observed response to localised heating

The dynamics of the system above, without the coupling of tracer to heating, has been much studied (e.g. Hoskins et al., 2003; Davies, 2015). The predicted response to an axisymmetric localised stationary heating is shown schematically in Figure 2($a$). A corresponding schematic diagram of the observed structure of anticyclonic vortices generated from aerosols from wildfires and volcanic eruptions, as discussed in §1 is shown in Figure 2($b$).

The structure shown in Figure 2($a$) may be explained briefly as follows. Consider a specified localised heating, with radial and vertical length scales respectively $L$ and $D$. Now consider the response to this heating, i.e. the right-hand sides of (4e), hence (5) and (7). In the context of (4e), part of the heating is balanced by the temperature tendency, $\partial_t T$, and part is balanced

by the term proportional to $w$, corresponding respectively to the responses in (5) and (7). According to (5), $\partial_t q$ is proportional to $\partial_z Q_s$, so in the situation where $Q_s$ is positive and localised, the response of the potential vorticity tendency is a negative anomaly above the heating and a positive anomaly below, with magnitudes estimated by (5) as $fRQ_s/DH_0N^2$. This will result in negative relative vorticity above the heating, positive relative vorticity below, and a positive temperature anomaly at the level of the heating with negative temperature anomalies above and below, all with magnitudes increasing linearly with time if $Q_s$ is kept constant. Simultaneously, according to (7), there is a secondary circulation response, with upwelling motion in the centre of the heating region, extending to levels above and below the heating, with compensating off-centre downwelling motion (indicated by arrows in Figure 2(a)). It is helpful to note further basic properties of the response to heating as captured by (5) and (7) and discussed in many previous papers. Firstly the responses in both $\psi$ (hence $T$ and $v$) and in $w$ to a localised heating extend away from that heating and the typical ratio of vertical to horizontal scales of the response is $f/N$, often known as Prandtl's ratio. Secondly the response to a distributed heating of the type shown in Figure 2(a) tends to be dominated, in the context of (4e), by $\partial_t T$, if the heating is shallow in the $f/N$-scaled sense (i.e. $D < fL/N$), and by the term proportional to $w$, if the heating is deep (i.e. $D > fL/N$). It is further helpful to note that in the deep case the shape of the vertical velocity tends to match that of the heating (the first term on the left-hand side of (7) balances the right-hand side), whereas in the shallow case the vertical velocity tends to be narrower than the heating (the second term on the left-hand side dominates the balance). The schematic Figure 2(a) depicts the case where the shape of the heating distribution roughly matches the $f/N$ scaling, with the vertical velocity partially balancing and somewhat narrower than the heating. Defining an aspect ratio $ND/fL$, it is useful to summarise this dependence on the aspect ratio as $w \sim c[ND/fL]RQ_s/N^2H_0$ where $c[ND/fL] \simeq (ND/fL)^2$ for the shallow case where $ND/fL \ll 1$, $c[ND/fL] \simeq \frac{1}{2}$ for the intermediate case where $ND/fL \sim O(1)$ and $c[ND/fL] \sim 1$ for the deep case where $ND/fL \gg 1$. The 'observed' structure (Figure 2(b)), in contrast, is a single-sign anticyclonic vorticity anomaly or correspondingly a negative PV anomaly, whose centroid is co-located with the centroid of the heating tracer, $\chi$. The azimuthal wind is correspondingly negative. The observed temperature dipole is a weakly stable configuration with a cold anomaly above the heating and a warm anomaly below, implying reduced static stability. The temperature and azimuthal wind are consistent with predictions for a single-sign potential vorticity anomaly (Bishop and Thorpe, 1994), however, the potential vorticity anomaly itself is different to theoretical predictions of the response to specified localised stationary heating as discussed above.

We hypothesize that this apparent difference can be resolved by incorporating the effect of the previously noted upwelling on the aerosol tracer and its associated heating, which will be displaced upward. The relevant problem to consider is not the response to a stationary localised heating, but the response to an ascending localised heating. As the heating arrives in a region

it will provide an anticyclonic PV forcing, and as it leaves it will provide a cancelling cyclonic PV forcing, suggesting that the response will be an anticyclonic PV anomaly moving with the heating.

The hypothesized behaviour can be illustrated by explicit calculation, (i) in a model in which the heating is simply specified as ascending at a given rate (§3.2) and then (ii) a model in which a heating tracer is transported by the secondary circulation (§3.3).

### 3.2 Upward moving heating

On the basis of the above qualitative discussion, before considering the full problem in which the tracer evolves according to (2) and hence determines the heating, it is useful to consider a new idealised problem in which the initial form of the tracer is specified and said tracer is subsequently assumed to move upwards at some specified velocity $W$ (denoted in capitals to distinguish it from $w$ which describes the actual vertical velocity in the secondary circulation induced by the heating).

The initial distribution of the tracer $\chi$ is the Gaussian profile specified in (8), with the same parameter choices as specified in §2.1, except that $z_c$ is taken to increase linearly in time consistent with the choice of $W$. The equations (4) are then integrated forward in time with $\chi$ and hence $Q_s$ as specified in §2. The evolution over a period of 13 weeks (91 days) is shown in Figure 3 with the choice $W = 3 \times 10^{-3} \mathrm{ms}^{-1}$. In the early evolution, shown in Figure 3($a$), the heating sets up a negative PV anomaly above and a positive PV anomaly below (consistent with §3.1 and Hoskins et al., 2003; Davies, 2015). Here, PV is calculated from the QGPV expression in square brackets on the left hand side of (5). The effect of the upward motion of the tracer, or equivalently the heating, visible at later times, is a negative vorticity anomaly moving with the region of tracer (Figure 3($b$)). The PV anomaly left behind the upward moving heating tracer is close to zero (Figure 3($c$)), since the negative forcing of PV as the heating arrives is cancelled by the positive forcing of PV as it leaves. Over time, the vertical location of the tracer maximum (red line) is increasingly co-located with the vertical location of minimum PV (black line), consistent with observations. Therefore the expected robust and persistent feature predicted by this model calculation is the ascending anticyclonic PV anomaly (and hence vorticity anomaly) that moves with the tracer.

However another prediction of this calculation is that, as a result of the early-time evolution, a positive PV anomaly is left below the initial region of heating (Figure 3($bc$)). This is not maintained by any forcing, so in the real atmospheric context its lifetime is likely to be limited by (i) deformation and mixing by the effects of the large-scale environmental flow (which are not captured in the axisymmetric model) and (ii) decay due to radiative damping; both these will be addressed to some extent in §3.4 and §3.5 respectively.

Further information on the response to a specified upward moving heating with zero thermal damping is given in Figure 4($ab$), which shows the time-height evolution of the temperature at the radial location of minimum temperature (typically

at $r = 0$). The temperature structure is consistent with that of the PV shown in Figure 3(c). PV anomalies correspond to vertical dipoles in the temperature, cold above warm for the anticyclonic PV anomaly and warm above cold for the cyclonic PV anomaly (with the persistence of the latter and its associated temperature structure expected to be unrealistic).

Motivated by the numerical solutions, we consider a steadily translating solution. We define a new coordinate $\xi = z - Wt$, such that the derivatives transform as

$$\frac{\partial}{\partial t} = \frac{\partial \xi}{\partial t} \frac{\partial}{\partial \xi} = -W \frac{\partial}{\partial \xi}, \qquad \frac{\partial}{\partial z} = \frac{\partial \xi}{\partial z} \frac{\partial}{\partial \xi} = \frac{\partial}{\partial \xi}, \tag{9}$$

and seek solutions where the variables depend on $\xi$ rather than on $z$ and $t$ separately. On substitution into the QGPV equation (5), neglecting any variation of $\rho_0$, replacing $q(r,z,t)$ and $\psi(r,z,t)$ by, respectively, $q_S(r,\xi)$ and $\psi_S(r,\xi)$ and integrating once with respect to $\xi$, we obtain,

$$q_S = \frac{1}{r} \frac{\partial}{\partial r} \left( r \frac{\partial \psi_S}{\partial r} \right) + \frac{f^2}{N^2} \frac{\partial^2 \psi_S}{\partial \xi^2} = -\frac{f}{W} \frac{RQ_s}{H_0 N^2} + \frac{\alpha}{W} \frac{f^2}{N^2} \frac{\partial \psi_S}{\partial \xi}, \tag{10}$$

to be solved with the boundary condition $\psi_S \to 0$ as $\xi \to \infty$. Figure 4(b) overlays $q_S$ (solid black contours) with the numerical solution $q$ (shading) at 50 days, the end of the simulation, demonstrating good correspondence between the negative PV anomaly and $q_S$ for no thermal damping.

The form of (10) shows that, in the absence of thermal damping, the magnitude of the vorticity anomaly/potential vorticity anomaly in the steadily translating solution is proportional to $W^{-1}$. This is because if the heating is of depth $D$ then at any level the heating is present for a time $D/W$ and correspondingly this is the time over which the potential vorticity forcing is felt. As has been previously noted, the magnitude of the potential vorticity forcing is, according to (5), $fQ_sR/DH_0N^2$ and therefore the magnitude of the resulting PV anomaly is $fQ_sR/WH_0N^2$.

In this idealised problem $W$ is simply imposed. However, (7) suggests that, again in the absence of thermal damping, the vertical velocity driven by a heating $Q_s$ with horizontal and vertical scales respectively $L$ and $D$ will be of order $c[ND/fL]RQ_s/N^2H_0$. This gives an estimate for $W$ as the heating-driven vertical velocity that moves the tracer and hence the heating distribution upward. Inserting this estimate into the above expression for the magnitude of the QGPV anomaly suggests that the magnitude of the QGPV anomaly will be $fQ_sR/H_0N^2 \times N^2H_0/c[ND/fL]RQ_s = f/c[ND/fL]$. However a further important point is that if $ND/fL$ is small (i.e., shallow heating) then the region of ascent tends to be narrower than that of the heating. Therefore a shallow region of tracer is unlikely to rise coherently as a result of its heating-induced vertical velocity. (This effect will be seen in the coupled tracer-dynamics calculations presented in the following section.) Therefore in practice a long-lived tracer distribution with $ND/fL$ small will not be possible, within this axisymmetric model.

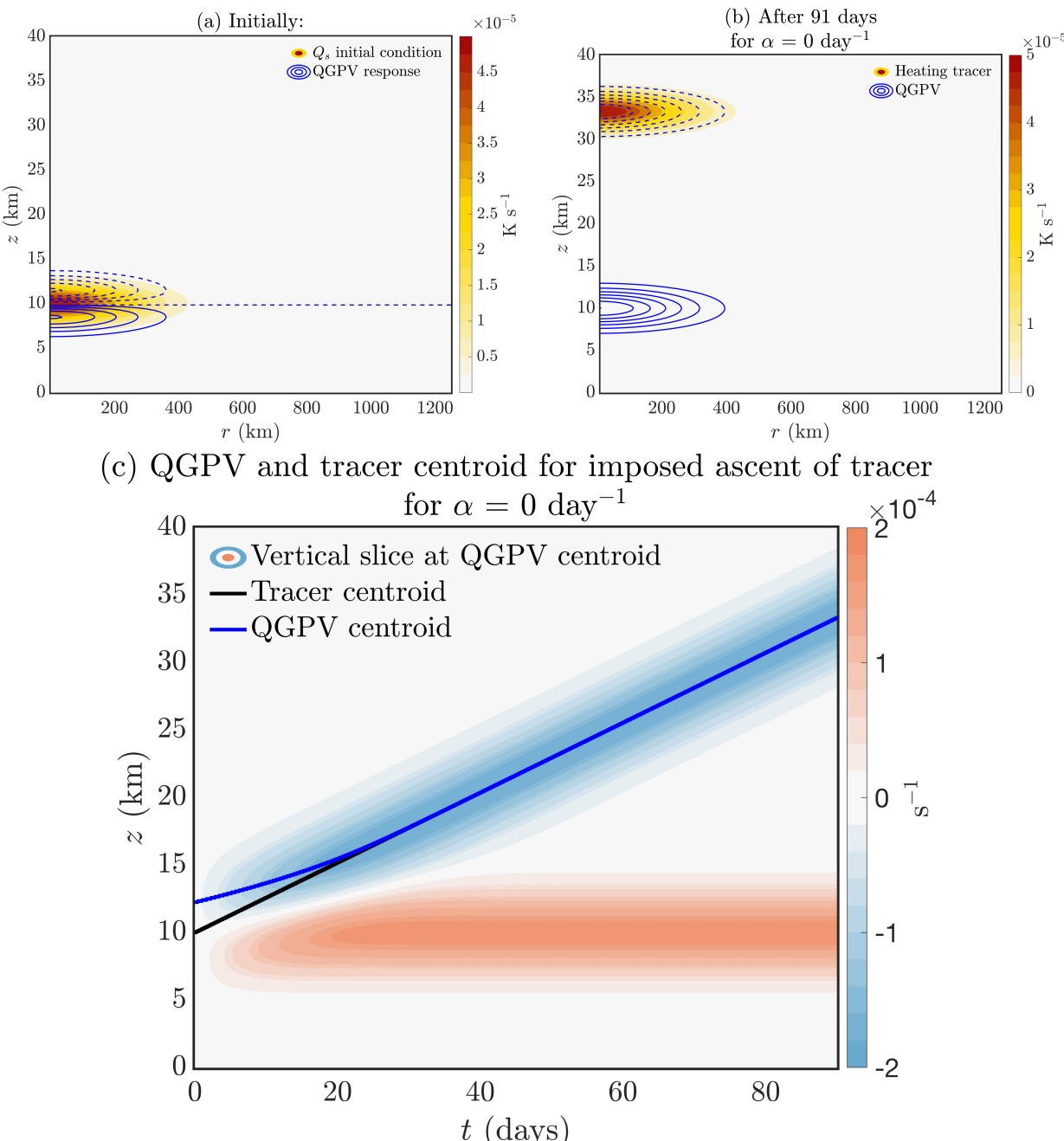

**Figure 3.** Tracer-filled vortex of radius 500 km, consistent with 1000 km vortex diameter detected from the Australian wildfires (Khaykin et al., 2020). The tracer (i.e. heating) is moved upward at $W = 3 \times 10^{-3} \mathrm{ms}^{-1}$. There is no thermal damping, i.e. $\alpha = 0$. Tracer (colour shading) with QGPV response (blue contours) is shown for (a) initial configuration (PV contour intervals are $5 \times 10^{-8} \mathrm{s}^{-1}$; note that this panel is a measure of QGPV tendency at initial time), and (b) after 91 days (PV contour intervals are $2.5 \times 10^{-5} \mathrm{s}^{-1}$). (c) Vertical slice of potential vorticity at the centroid of negative PV. The altitude of the tracer centroid (red line) and anticyclone centroid (black line) are overlaid.

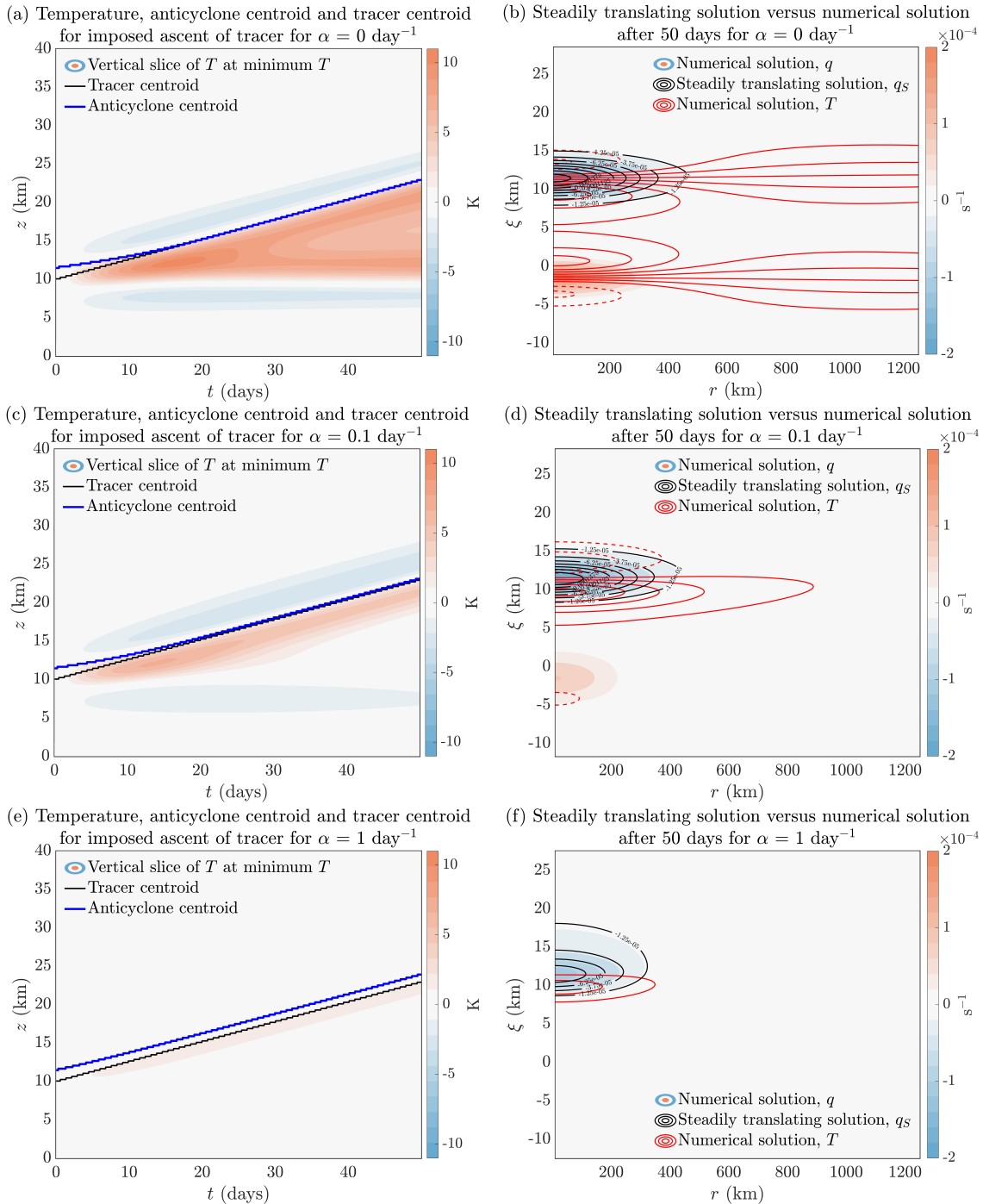

**Figure 4.** Evolution of temperature and QGPV for steadily upward moving heating, including influence of thermal damping: [first row] no thermal damping, $\alpha = 0\,\mathrm{day}^{-1}$, [second row] $\alpha = 0.1\,\mathrm{day}^{-1}$, [third row] $\alpha = 1\,\mathrm{day}^{-1}$. [first column] Same as panel Figure 3(c), but for temperature. [second column] Steadily translating solution for QGPV, $q_S$ (black contours) from (10), at 50 days underlaid with numerical solution (coloured shading). Temperature contours are overlaid (contour interval 1K, dashed for negative and solid for positive temperatures).

Thermal damping, through radiative transfer, is expected to modify the response to the forcing of PV by the heating and hence to modify other dynamical quantities including temperature. This can be investigated by including non-zero values of $\alpha$ in the calculation. Figure 4($df$) shows good correspondence between the steadily translating solution and the initial value calculation for (d) $\alpha = 0.1\mathrm{day}^{-1}$ and (f) $\alpha = 1\mathrm{day}^{-1}$. Lestrelin et al. (2021) on the basis of assimilation increments for temperature estimate a radiative damping timescale of 6-7 days, which is used by Podglajen et al. (2024). This estimate is consistent, given the vertical length scales, with scale-dependent calculations in Haynes and Ward (1993). (Note that textbook estimates of radiative damping timescales in the lower stratosphere are usually significantly longer, but typically assume deep vertical structures.)

The results shown in Figure 4($bdf$) are from the initial-value calculation, but they are consistent with the steadily translating solution of (10) except that, as expected, the low-level cyclonic vortex is absent in the steadily translating solution because the initial conditions have been forgotten.

For this simple experiment where the tracer is moved upwards at constant speed, the thermal damping has two major effects. The first is that, as can be seen in the right-hand panels, it dissipates the low-level cyclonic PV anomaly, which is not being maintained by any forcing. As described by Haynes and Ward (1993), the effect of thermal damping is to reduce the magnitude of the PV anomaly but also to change its shape, because the thermal damping acts on temperature anomalies and not on velocity anomalies. This change of shape can be seen in the two lower right-hand panels. The second effect, perhaps more important in the realistic context, is that it alters the structure of the other dynamical fields accompanying the upward moving heating. Thus it can be seen that the magnitude of the upward moving temperature anomalies at $r = 0$ are reduced as $\alpha$ increases. The effect on the magnitude of the potential vorticity anomaly is much weaker as the thermal damping acts only on the temperature field, though some effect can be seen in the $\alpha = 1\mathrm{day}^{-1}$ case.

To consider the extent to which the thermal damping affects the dynamics, and in particular the size of the potential vorticity anomaly, it is necessary to consider the size of the second term on the right-hand side of (10) relative to the left-hand side. The ratio of these terms is $(\alpha D/W) \times c[ND/fL] \times f^2 L^2/N^2 D^2$ where the function $c[\cdot]$ has been defined previously. This is $\alpha D/W$ multiplied by a factor that is small for deep heating and close to 1 for shallow heating. $\alpha D/W$ is the ratio of the time taken for the heating to move through its own depth to the thermal damping time. The further factor dependent on the aspect ratio $ND/fL$ means that for the thermal damping term to be important the ratio $\alpha D/W$ has to be quite large, unless the heating distribution is shallow. This explains why the effect of the thermal damping on the potential vorticity anomaly is apparent in Figure 4 only for $\alpha = 1\mathrm{day}^{-1}$.

To assess the effect of thermal damping on the vertical velocity, first note from the argument above that if $(\alpha D/W) \times c[ND/fL] \times f^2L^2/N^2D^2 \gg 1$ then the dominant balance in (10) implies in turn that the two terms on the right-hand side of (7) will tend to cancel and hence that the vertical velocity will be much smaller than that estimated for $\alpha = 0$. Therefore a simple conclusion is that the scaling arguments above do not apply if $W \ll \alpha D \times c[ND/fL] \times f^2L^2/N^2D^2$, or equivalently if $RQ_s/N^2H_0 \ll \alpha D \times f^2L^2/N^2D^2$ essentially setting a minimum value for $Q_s$ for this simple 'self-lofting' tracer model to be valid.

Explicit calculation indeed shows that, for a fixed $Q_s$, the vertical velocity $w$ at the centre of the tracer distribution tends to reduce as $\alpha$ increases. However the level of the maximum vertical velocity shifts upwards, so that, for modest values of $\alpha$ there is significant increase in $w$ in the upper part of the tracer distribution.

There are two key conclusions from the above discussion. The first is that the magnitude of the quasigeostrophic PV anomaly, in this steadily translating case, is independent of $Q_s$. The second is that the relevant estimate for the magnitude of said QGPV anomaly is $O(f)$ for intermediate or deep (i.e., $ND > fL$) vortices, as the aspect ratio factor $c[ND/fL]$ is 1/2 or 1 respectively. (Typical values for the Australian wildfires' main vortex suggest $ND > fL$.) The prediction of $f$ as the magnitude of the QGPV anomaly implies that the anticyclone has Rossby number $Ro \sim 1$, i.e. the anticyclone is long-lived, as typical external shear or strain rates would be a fraction of $f$. Hence, as the magnitude of the peak vorticity increases over time, the likelihood of survival of the vortex depends on the early stages of its evolution: if the peak vorticity increases rapidly, the vortex is much more likely to persist. Conversely, if the peak vorticity increases slowly, there is a high likelihood of its disruption by the background shear or strain. Furthermore, given that the predicted peak vorticity is $O(f)$ it has to be accepted that quasigeostrophic theory may be inadequate to describe some aspects of the evolution. The practicalities of solving the non-QG Eliassen problem were noted in §2.1. In summary then, the above scaling arguments, which for self-consistency require a minimum value of $Q_s$, suggest that the maximum magnitude of the PV anomaly, and correspondingly of the relative vorticity and of the azimuthal velocity, are independent of the magnitude of the heating but the ascent rate of the PV anomaly is not.

## 3.3  Coupled aerosol-dynamics system

We now solve the full quasigeostrophic equations where the tracer equation is solved explicitly and determines the heating. The secondary circulation is solved to find $w$ from (7) and hence $u$ from (4c); these velocities then advect the heating tracer in (2).

To probe how the vortex evolution depends on the initial aspect ratio of the tracer bubble, we explore three different initial conditions in Figure 5: (first row) the 'standard' initial condition shown in Figure 3(a) with parameters given in (8), (second row) the deep initial condition, where $r_0$ is 2/3× and $z_0$ is 3/2× their values in the standard case, (third row) the shallow initial

condition, where $r_0$ is $3/2\times$ and $z_0$ is $2/3\times$ their values in the standard case. Note that the scaled aspect ratios $Nz_0/fr_0$ are respectively $1.5$, $3.4$ and $0.67$ for the standard, deep and shallow cases. At early time, the localised tracer (i.e. heating) implies anticyclonic circulation above, cyclonic below. As time goes on, self-lofting of the localised tracer (and hence heating) occurs, with peak vertical velocities at $\approx 3.5\times 10^{-3}\mathrm{ms}^{-1}$ (typical values of $u$ as calculated from continuity ($4c$) are of order $0.1\mathrm{ms}^{-1}$).

In all three cases, the maximum tracer abundances occur at the top of the tracer structure, which is consistent with detected aerosol bubbles following the Australian and Canadian wildfires (Khaykin et al., 2020; Lestrelin et al., 2021). Podglajen et al. (2024) highlighted this feature, which also emerges as a prediction of our model because $\partial_z w < 0$ at the upper boundary of the tracer structure. Though subject to uncertainty, minimum PV values from ERA5 are generally at the centre of the anticyclones that accompany the tracer structures (whereas in our solutions the tracer structure tends to deepen with time extending well below the anticyclone, as they do in Podglajen et al. (2024)). That said, the *centroid* of the tracer structure lies within the anticyclone (right column of Figure 5), as does the region between the induced cold and warm anomaly, a feature that is also in agreement with composites of vorticity and temperature (Figure 6 in Khaykin et al. (2020) and Figure 8 in Lestrelin et al. (2021)). The QGPV values are seen to approach $O(f)$ over time, in agreement with scaling arguments in §3.2, although those arguments were based on the observed contained aerosol bubble and hence do not account for the tracer tail seen in Figure 5.

The precise relation between the tracer distribution (hence the heating distribution) and the vertical velocity distribution is expressed by (7). As noted previously, the distribution of $w$ tends to be narrower and taller than the tracer distribution if the latter is shallow (in the $f/N$-scaled sense) and broader and shallower than the tracer distribution if the latter is deep. Therefore with a deep initial tracer distribution most of the tracer will be effectively transported upwards, whereas with a shallow initial tracer distribution it will be the central part (in the horizontal) that is most effectively transported upward. This effect is visible in Figure 5 (left column). For the shallower initial conditions the tracer at larger radius is left behind the central part. For the deep initial condition the majority of the tracer moves upwards as one structure. As noted previously, the vertical velocity is larger when the tracer distribution is deep. This effect may be seen in the early-time upward motion of the tracer centroid seen in Figure 5 (black lines in right column). The initial ascent rate is largest for the deepest case and smallest for the shallowest case. At later times the shape of the tracer distribution changes which affects the ascent rate. The shapes of the central region of the tracer distribution are more similar between the cases at later times than they are initially, therefore the later-time ascent rates are more similar.

We now turn our attention to the shape of the potential vorticity anomalies, which are forced by the heating and are therefore determined by the time history of the tracer. As expected from the simulations with specified upward motion of the heating, the dominant features are an anticyclonic potential vorticity anomaly moving upward with the tracer and a cyclonic potential

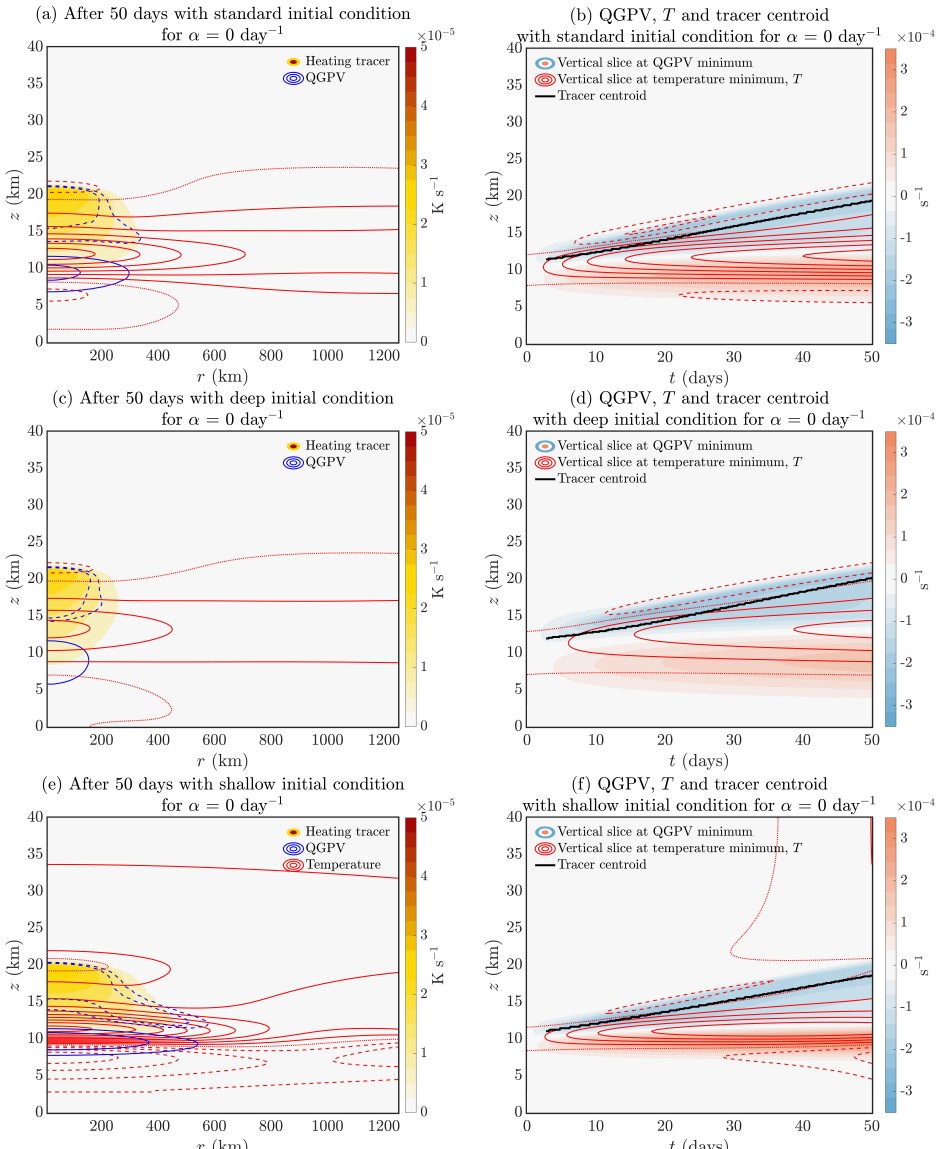

**Figure 5.** Evolution of a tracer-filled vortex, where the tracer generates heating and its own secondary circulation. No radiative damping has been included ($\alpha = 0$ day$^{-1}$) and horizontal and vertical diffusivities are $\kappa_h = 10^3\,\mathrm{m}^2\mathrm{s}^{-1}$ and $\kappa_v = 2\times10^{-1}\,\mathrm{m}^2\mathrm{s}^{-1}$ respectively. (first row) The (standard) initial condition of $\chi = Q_s$ is (8), where values of $\chi_0, r_0, z_0, z_c$ values are specified, i.e. the same as shown in Figure 3. (second row) Deep initial condition, where $r_0$ is $2/3\times$ and $z_0$ is $3/2\times$ their values in Figure 3(a). (third row) Shallow initial condition, where $r_0$ is $3/2\times$ and $z_0$ is $2/3\times$ their values in Figure 3(a). The scaled aspect ratios $Nz_0/fr_0$ are 1.5, 3.4 and 0.67 for the standard, deep and shallow cases respectively. (ace) After 50 days, the tracer field (colour shading), potential vorticity (blue contours with interval $10^{-4}\mathrm{s}^{-1}$ with an added contour at $-0.75 \times 10^{-4}\mathrm{s}^{-1}$ to indicate the weak anticyclonic tail), and temperature (red contours with interval 2.5 K); dashed lines are negative values and solid lines are positive values. The dotted red line is the zero $T$ contour. (bdf) Vertical slice of QGPV at QGPV minimum (coloured shading) and temperature (red contours with interval 2.5K for (bd) and 5K for (f) for readability, with same dashed, solid, dotted convention as (ace)), with the vertical location of the tracer centroid plotted in black solid line.

vorticity anomaly left at a fixed level below and essentially determined by the initial distribution of the tracer. The fact that for the shallower initial conditions there is a central portion of the tracer distribution that is ascending more rapidly and an outer part that is ascending less rapidly implies similar geometry for the anticyclonic potential vorticity anomaly. The central part of the potential vorticity anomaly is largely forced by the central part of the tracer distribution and ascends with it. The outer part of the potential vorticity anomaly is largely forced by the outer part of the tracer distribution and ascends with it, more slowly than the central part. This effect is not seen for the deep initial condition because the entire tracer distribution moves upwards together.

We now consider the role of thermal damping in the fully coupled model. While the conclusions from simple scaling arguments we put forth in §3.2 that arose from the imposed ascent case would be expected to hold in certain situations, the behaviour seen in the fully coupled model is more likely to describe strong aerosol-filled vortices. Tracer, vertical velocity, and potential vorticity profiles after 50 days are shown in Figure 6 for the case of (a) $\alpha = 0.1$ day$^{-1}$ and (b) $\alpha = 1$ day$^{-1}$ (time integrations are from the standard initial condition). On comparing Figure 5($a$) to Figure 6, the main effect that thermal damping introduces is to increase the rate of ascent of the leading edge of the tracer structure (as originally suggested by Podglajen et al. (2024) who report an increased vertical ascent rate of the tracer front due to a reduced decay of maximum tracer concentrations), forming a deeper more radially compact shape. In our model, this is because the effect of thermal damping on $w$ is positive in the top part of the tracer structure (not shown). With thermal damping, the accompanying anticyclonic anomaly is correspondingly more radially compact, and the cyclonic PV anomaly, temperature, and azimuthal velocity have reduced magnitude and radial extent.

As has been noted in the previous section, neither the cyclonic potential vorticity anomaly nor its associated dipole temperature signal have been emphasised in the literature, though there is some indication of such an anomaly in the modelling study by Doglioni et al. (2022) (Figures 5(a) and 6(a).) We suggest that the cyclonic PV persists in our model because the axisymmetric framework omits important 3D processes.

### 3.4 Solutions with 'vortex-stripping' adjustment for the tracer field

The details of the coupled tracer-dynamics structure discussed in the preceding section are significantly different from those that have typically been observed, such as the persistence of anomalies at lower levels, the cyclonic PV anomaly and the trailing tracer features and accompanying PV anomalies that are particularly prominent for the standard and shallow initial conditions. That said, it is also the case that the regions of substantial tracer concentration extend well below the anticyclonic PV anomaly in the central part of the upward moving structure, and this applies even in the case of the deep initial condition. These differences can be attributed to our model being axisymmetric and hence not capturing 3D effects such as vortex stripping (referred to

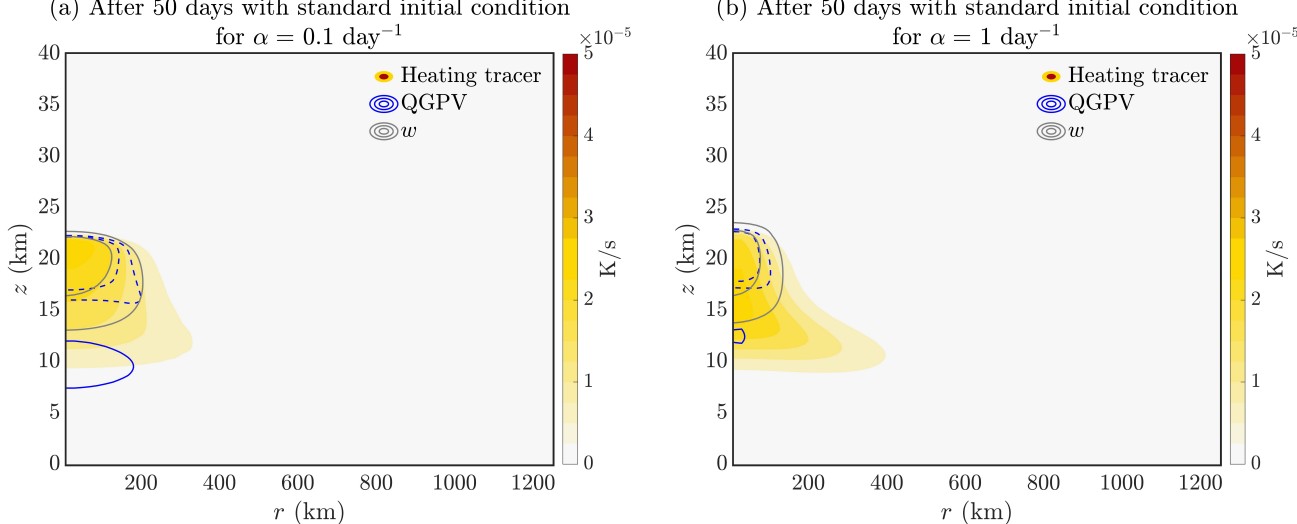

**Figure 6.** Full quasigeostrophic model integrated from the standard initial condition for 50 days with thermal damping for: (a) $\alpha = 0.1$ day$^{-1}$, (b) $\alpha = 1$ day$^{-1}$. Solutions at 50 days are shown for the heating rate (i.e., tracer) in filled contours, vertical velocity in grey (contour interval $1 \times 10^{-3}$ms$^{-1}$), quasigeostrophic PV in blue (contour interval $10^{-4}$s$^{-1}$ with an added contour at $-0.75 \times 10^{-4}$s$^{-1}$, consistent with Figure 5(*ace*)).

here as generally resulting from a combination of horizontal and vertical shear), which allow the vorticity distribution to have a very direct effect on, for example, tracer dispersion. These missing effects can be incorporated in a very simple ad hoc way in the axisymmetric model by incorporating an adjustment to the tracer field whereby any tracer lying in regions where the PV or vorticity anomaly is less than a critical threshold is instantaneously removed. The justification is that in reality tracer outside of coherent vortices will be rapidly mixed. To focus on the dynamics of the persistent anticyclonic vortices, rather than on

their initial formation, the adjustment is applied at an intermediate time, when, within the interactive tracer-dynamics model, the regions of anticyclonic PV are already significantly displaced from the region where tracer was initially concentrated and, furthermore, the tracer is retained only in anticyclonic regions. The simulations reported in the previous section were repeated and the adjustment applied only after 14 days, but applied continuously after that time. The criterion for retaining (or removing) tracer could be varied and in the illustrative cases to be shown was chosen on the basis of PV being less than (or greater than)

the value $q_{\text{crit}} = -10^{-5}$ s$^{-1}$. (A vorticity- rather than PV-based threshold can be chosen but the results are very similar.)

Figure 7 shows solutions for the standard initial condition (panel (*a*)) without and (panel (*b*)) with tracer adjustment. The effect of removal of tracer lying outside the critical PV contour after 14 days is immediately apparent. The effect of the tracer adjustment on the PV can be seen on comparing panel (*c*) to Figure 5(*a*) and on comparing panel (*d*) with Figure 5(*b*). The

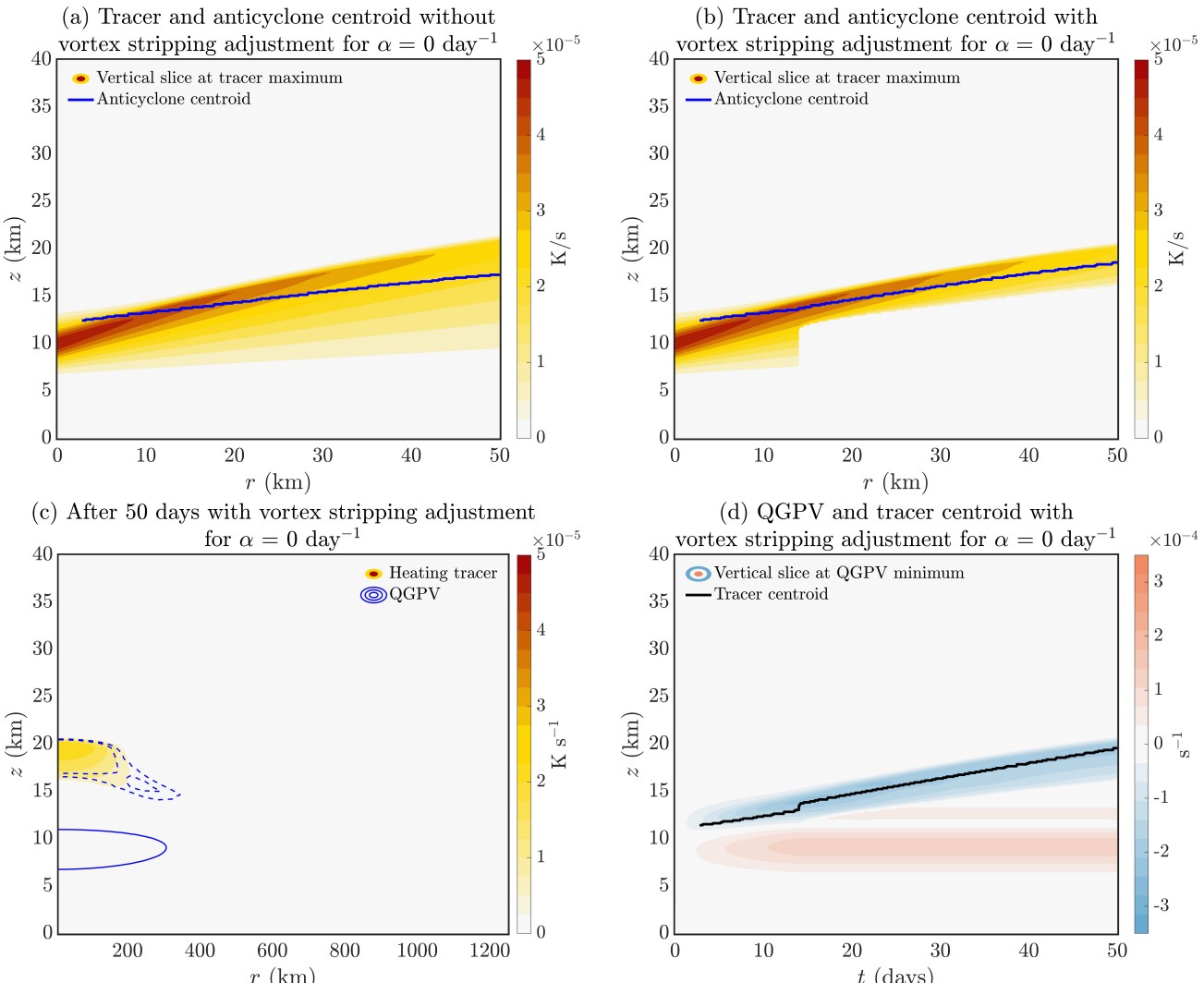

**Figure 7.** Vertical slice of (ab) tracer abundance at the tracer maximum and (d) QGPV at QGPV minimum (a) without and (bd) with vortex-stripping adjustment. (c) Cross-section of tracer (colour contours) and QGPV (blue contours with interval $10^{-4}\mathrm{s}^{-1}$; dashed lines are negative values and solid lines are positive values; an additional contour at $-0.75 \times 10^{-4}\mathrm{s}^{-1}$ has been added to indicate the region of weak anticyclonic PV) after 50 days with vortex-stripping adjustment. When the adjustment is applied, tracer abundances in $q > -10^{-5}\ \mathrm{s}^{-1}$ regions are set to zero. Here, solutions are for the standard initial condition shown in Figure 3(a).

upward propagating anticyclone is shallower, with sharper vertical gradients of PV. The tail-like structure seen in panel (c) is formed of PV generated at early times before the vortex stripping adjustment begins. There is also an effect on the low level cyclonic anomaly, implying that in the case without tracer adjustment the trailing structures in the tracer field are playing a role in maintaining this lower level anomaly in PV.

The abrupt adjustment of the tracer field at 14 days is of course unrealistic. Furthermore the fact that the coincidence between the tracer field and the anticyclonic vorticity is greater with the adjustment than without it is a direct consequence of the adjustment and therefore by itself not very significant. The important point that this calculation illustrates is that the coherent upward-propagating tracer-vortex structure is robust to the inclusion of the adjustment, giving greater confidence that the mechanisms described here are viable in the real atmosphere. Careful comparison of Figure 7($a$) and ($b$) shows that the adjustment has only a small effect on the upward propagation of the structure. In the non-adjusted case the tracer and vorticity maxima have reached a height of 22km after 50 days as compared with 21km in the non-adjusted case. It may also be seen that the adjustment gives only a small change to tracer concentrations in the central part of the structure. This is consistent with the prediction of our scaling arguments (presented in §3.2) that the rate of ascent is determined primarily by typical tracer, and hence heating, values within the structure.

### 3.5  Effects of vertically varying density

Thus far, by using a very large value for $H$, the density $\rho_0$ is roughly constant, equivalent to making the Boussinesq approximation. In this section, we will explore the influence of non-constant density by solving the full quasigeostrophic equation with tracer adjustment, and choosing $H = H_0 = 7000$m so that density now varies substantially with $z$. Note that the variation of density over the initial depth of the tracer distribution is relatively small. It is the effect of density variation as experienced by the upward moving tracer and accompanying dynamical anomalies that is of interest. We have considered several simulations with this choice of $H$ and for illustrative purposes we show only one set in Figure 8, with non-zero thermal damping $\alpha = 0.1$ day$^{-1}$ and with tracer adjustment as discussed in the previous section. Solutions with $T_B$ varying linearly (so that buoyancy frequency increases with $z$) behave similarly to those in Figure 8. We focus on the effect of varying $\rho_0$ in this subsection.

The variation of $\rho_0$ has an effect on the tracer transport via (2) and changes the operators acting on $\psi$ in (5) and on $w$ in (7), such that anomalies resulting from a localised $Q_s$ tend to extend further above the region of non-zero $Q_s$ than they do below. The net effects of these differences on the evolution can be seen in Figure 8, comparing panels ($b$), ($e$) and ($f$), with varying $\rho_0$ to panels ($a$), ($c$) and ($d$), with constant $\rho_0$. In the former case there is a larger tracer bubble, tracer concentrations maintained at higher values for longer and larger ascent rates. A possible explanation is that as the tracer bubble ascends and enters less dense air, its volume increases to respect mass conservation. Whilst this by itself does not directly affect concentrations and hence ascent rates, it will imply larger spatial scales and a weaker effect of diffusion. There is hence less diffusive reduction in tracer concentrations, resulting in larger ascent rates and anomalies in dynamical variables that are large in amplitude and larger in scale, consistent with the modelled evolution.

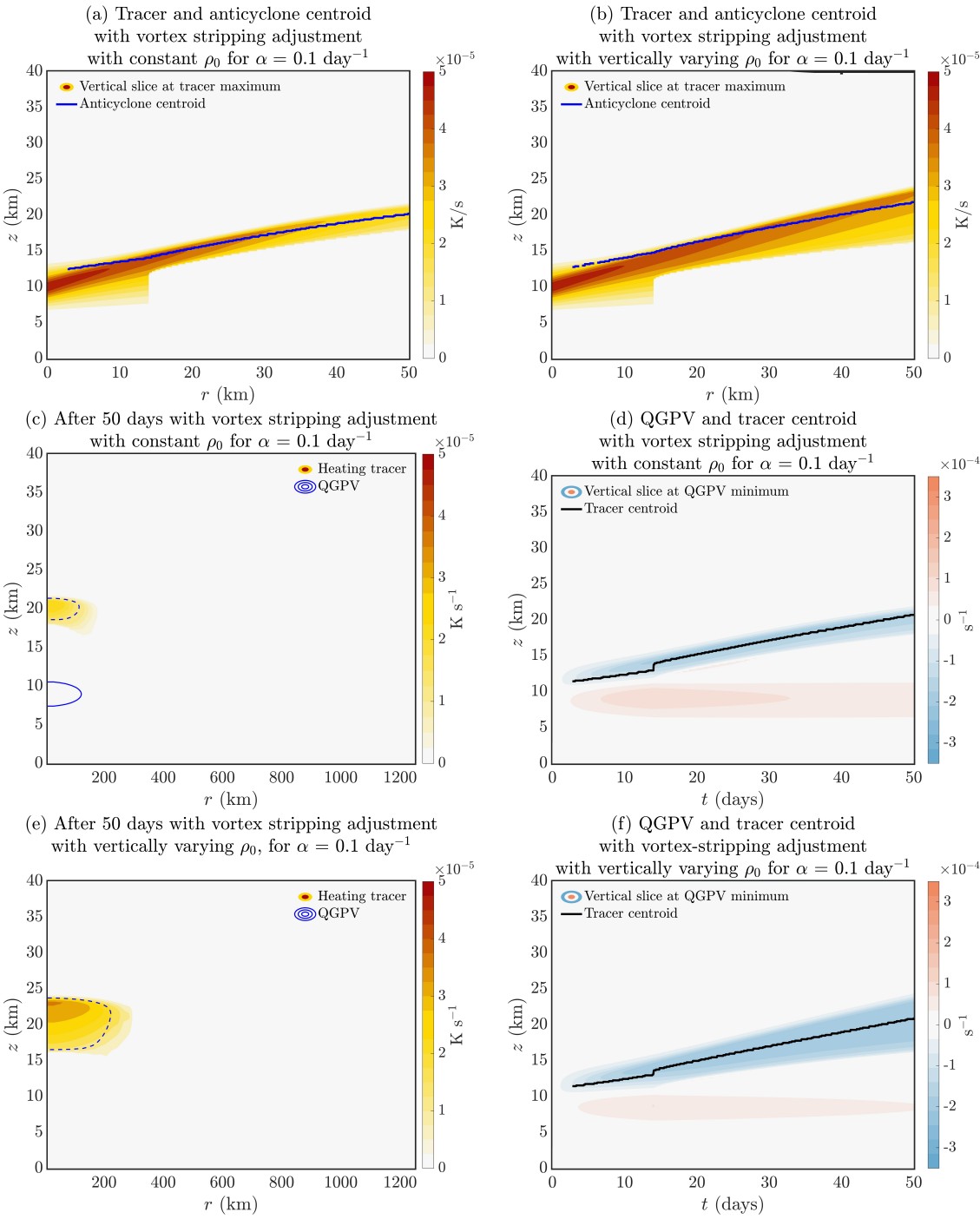

**Figure 8.** Full quasigeostrophic model with vortex stripping adjustment, $\alpha = 0.1 \text{day}^{-1}$ and with (abe) constant $\rho_0$ and (cdf) vertically varying $\rho_0$. Vertical slices of (ab) tracer at tracer maximum, (df) QGPV at QGPV minimum (coloured shading) with the vertical location of anticyclone centroid and tracer centroid plotted in blue and black lines respectively. (ce) Tracer field after 50 days (colour shading) with potential vorticity contour overlaid (blue dashed for negative PV with interval $10^{-4}\text{s}^{-1}$).

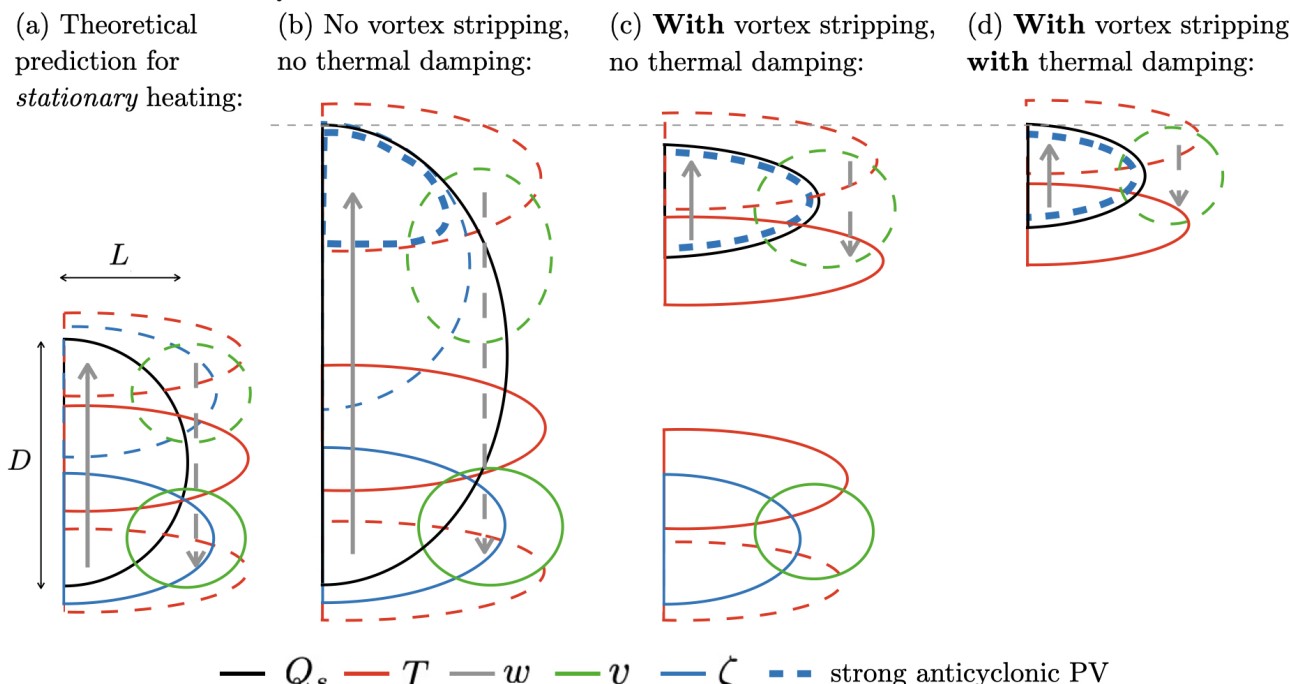

**Figure 9.** Schematic summarising the physical processes in our theoretical model for aerosol-filled vortices. For reference, panel (a) displays the theoretical prediction for stationary heating (e.g., Hoskins et al., 2003; Davies, 2015). The key behaviour of our coupled tracer-dynamics system is shown in (b) without vortex stripping and without thermal damping, (c) when vortex stripping is applied without thermal damping, (d) with both vortex stripping and thermal damping incorporated.

A further general point about non-Boussinesq effects suggested by these solutions is that the decreasing density tends to offset the effect on ascent rates of tracer leakage from the upward moving bubble. This may be an important effect in the real atmosphere where the tracer bubbles may ascend around 15km, equivalent to a factor of 10 reduction in density. Said differently, if the tracer bubble was not leaking, the non-Boussinesq effects would act to increase the volume of the bubble and hence the ascent rate via the diffusive argument made above. This increased bubble volume and speedup of vortex ascent is not detected in the observed cases on record thus far, suggesting that the tracer bubbles leak material, consistent with existing observational evidence (e.g. Khaykin et al., 2020).

We now present Figure 9 as a schematic representation of the key findings from our simple model. For reference, panel (*a*) shows the theoretical prediction for stationary heating (e.g., Hoskins et al., 2003; Davies, 2015) with a Gaussian heating tracer profile that generates upwards vertical velocity, anticyclonic QGPV above and cyclonic QGPV below, with corresponding

temperature and wind fields. Panel (*b*) shows the key behaviour of our tracer-vortex system without vortex stripping nor thermal damping, corresponding to Figure 5(*a*). The heating tracer has self-lofted with an accompanying anticyclonic PV anomaly that is strongest at the top of the tracer structure, with a corresponding cold temperature anomaly above and warm temperature anomaly below, and weaker tail at the top of the outer trailing tracer regions. As the tracer ascends it leaves behind it a cyclonic PV anomaly with a warm temperature anomaly above and a cold temperature anomaly below. Panel (*c*) shows the behaviour when our vortex stripping adjustment is applied, corresponding to Figure 7(*c*). The main effects of the adjustment are a shallower anticyclone and weaker lower-level cyclonic PV, temperature, and wind anomalies. Panel (*d*) shows the behaviour of our model with both vortex stripping and thermal damping, corresponding to Figure 8(*c*). The tracer bubble and accompanying anticyclone have more compact structures and the lower-level cyclone and temperature and azimuthal wind anomalies have been damped out.

### 3.6   Perspective on Podglajen et al. (2024)

As we submitted this paper we became aware that an independent paper on the dynamics of heating-driven vortices had been submitted for publication elsewhere (Podglajen et al., 2024, henceforth P2024). While revising our manuscript, we added this subsection to provide a perspective on the two studies: their similarities and points of difference, aspects unique to each, and limitations.

The key similarities between the two studies are, first, the PV-based description of the dynamics and, second, the introduction of an active tracer representing sunlight-absorbing aerosols whose mixing ratio determines the shortwave heating rate, such that, $Q_s \propto \chi$. Both studies predict that the self-lofting effect of this tracer implies an upward moving heating, generating anticyclonic PV above and cyclonic PV below, resulting in an anticyclone that moves upward with the tracer and a cyclone that is left behind at low levels. P2024 identify that thermal damping tends to increase the ascent rate of the tracer and we have verified that this also applies to our model.

One significant difference between the two studies is in the initial condition. We consider an initial condition where there is a specified tracer anomaly and the flow is at rest. P2024, on the other hand, consider an initial condition with and without an anticyclone present, together with the tracer anomaly and report that the coherence and ascent of the tracer bubble is enhanced with the initial anticyclone. We will discuss this point further in §5.

The important methodological differences between P2024 and our study are as follows. P2024 use both a simplified axisymmetric model and a full 3-D non-hydrostatic numerical model. In their axisymmetric model formulation, P2024 adapt coordinate transformations previously applied in tropical cyclone modeling approaches (Schubert and Alworth, 1987; Schubert, 2018) to express the evolution in terms of 1-D nonlinear wave equations with potential temperature as the spatial coordinate.

The radial coordinate is the potential radius (Schubert and Hack, 1983). These equations represent the upward movement of the tracer and predict the formation of a finite-time singularity associated with steepening vertical tracer gradients and PV values approaching zero. The singularity may be resolved by including a vertical tracer diffusion. P2024 additionally include the effect of thermal damping in the 1-D model, which requires the heuristic application of a QG balance condition. The 1-D model results provide a valuable basis for interpreting the 3-D model simulations. The latter show, in integrating from an
axisymmetric initial condition, that axisymmetry is largely preserved.

The strong advantages of the P2024 approach are that the use of the numerical model avoids the technical difficulties of integrating balanced equations when PV becomes near-zero and that the 1-D model gives clear insight into some of the mechanisms that are operating in the numerical model. Therefore P2024 are able to provide a self-consistent prediction that PV values become very small in the anticyclone. Our scaling arguments and model simulations have similarly predicted that PV
values in the anticyclone are small, following from $|q| \sim |\zeta| \sim f$ and noting from Berrisford et al. (1993) that an approximate form of the full PV $P$ is $P \approx -g/\mathrm{d}_\theta p_0(f + q)$, where $p_0$ is a function of the vertical coordinate (which can be transformed to the coordinate system of choice).

A particular shortcoming of QG dynamics is that the equations neglect vertical advection of PV and the evolution of the QGPV (5) is determined solely by the diabatic forcing term proportional to the vertical gradient of the heating. Investigation
of the P2024 model, in particular their equation (17), suggests that it is the diabatic forcing term that plays the dominant role in reducing the PV in the leading edge of the tracer cloud and neglect of the advective term in the PV equation makes only a minor difference. However the vertical advection, in that particular case, plays a larger role in the deepening of the lower level cyclonic PV anomaly below the tracer cloud suggesting that, more generally, the QG model may not capture the detail of the cyclonic PV anomaly correctly. However one could also remark, again with the caution that re-analysis PV may miss
important details, that a cyclonic PV anomaly below the ascending tracer bubble has not been evident in the descriptions of observed cases (Khaykin et al., 2020; Kablick III et al., 2020; Lestrelin et al., 2021).

Whilst accepting that the calculations of our study are subject to the limitations of QG theory whereas P2024's are not, there are ingredients of the problem that might allow the QG model we have presented to be more useful than might appear. The first is that, the scaling arguments we lay out in this study match estimates from observations of, e.g., relative vorticity, and with
600 suitable modification of parameters can be extended beyond quasigeostrophy (e.g., see equation (33b) in Hoskins et al., 1985). Second, in the 3-D simulations of P2024, the central portion of the anticyclone appears to have PV that is small, but not zero, while the flanks have near-zero PV. This is different to the anticyclonic PV pattern seen in reanalysis (subject to uncertainty), where the vorticity is near-zero in the central region of the anticyclone. Third, the tracer structures presented by P2024 (and

our study) tend to extend upwards with time rather than, as is seen in the aerosol observations, move upwards as a compact structure (e.g., Figure 5(a) in Khaykin et al., 2020). Therefore the structure of the trailing cyclone, which we have noted above may be poorly captured by QG, may simply be an artificial aspect of these axisymmetric models. As has been noted previously, part of the reason for the contained tracer structure may be the action of the flow external to the tracer bubble, which may have the effect not only of removing tracer but also changing the dynamics.

## 4 Formation and self-organisation of heating-driven coherent vortices

Thus far, we have explored the dynamics of a vortex evolving from an initial condition of a localised bubble of heating tracer as described by an initial condition (8). However, this neglects the question of how and under what conditions such a localised bubble of tracer may form in the first place. We now consider the question of how an initially horizontal homogeneous layer of heating tracer, subject to small perturbations will naturally organise itself into localised structures (such as plumes and, perhaps, bubbles) initially through a linear instability.

To study this problem without requiring a full three-dimensional numerical simulation, it is helpful to assume that the configuration is two-dimensional, depending only on the two Cartesian coordinates $x$ (horizontal) and $z$. The evolution equations used previously for dynamical variables and for the tracer distribution require some minor modification to take account of the two-dimensional rather than axisymmetric geometry.

Adopting the QG framework, the required modified form of (5) is

$$\frac{\partial}{\partial t}\left[\frac{\partial^2 \psi}{\partial x^2} + \frac{1}{\rho_0}\frac{\partial}{\partial z}\left(\frac{f^2}{N^2}\rho_0\frac{\partial \psi}{\partial z}\right)\right] = \frac{f}{\rho_0}\frac{\partial}{\partial z}\left(\frac{\rho_0 R Q_s}{H_0 N^2}\right) - \frac{\alpha}{\rho_0}\frac{\partial}{\partial z}\left(\frac{\rho_0 f^2}{N^2}\frac{\partial \psi}{\partial z}\right). \tag{11}$$

It is useful to note the QG form of the thermodynamic equation $(4e)$,

$$\frac{\partial}{\partial t}\left(\frac{\partial \psi}{\partial z}\right) + \frac{N^2}{f}w = Q_s - \alpha\frac{\partial \psi}{\partial z} \tag{12}$$

where $\alpha$, $f$ and $N$ are as defined previously. (The Cartesian form of the equation (7) for $w$ may be obtained by combining (11) and (12).) It is convenient in what follows to assume that $N$ is constant and to neglect vertical variation of $\rho_0$. These assumptions could be relaxed if needed. Note that the assumption of two-dimensionality means that, as in the axisymmetric case, no advection by the geostrophic velocities appears in these equations. (The geostrophic velocity is purely in the $y$-direction; in 2-D, there is no variation in this direction.)

The tracer abundance, $\chi$, is assumed directly proportional to the heating, $Q_s$, as previous, and satisfies (2), with $r$ replaced by $x$, $(u,w)$ interpreted as the ageostrophic velocity field in the $(x,z)$ plane and the geometric factors $r^{-1}$ and $r$ appearing in the horizontal diffusive term on the right-hand side omitted.

## 4.1 Linear stability analysis

We now consider the configuration where the tracer abundance (and hence the heating) is a function of $z$ only, such that, $\chi = \chi_0(z)$, $Q_s = Q_{s0}(z)$. The steady state solution is given by a balance between the anomalous heating from the aerosol and radiative damping, i.e. $\alpha \, d_z \psi_0 = Q_{s0}$. We then consider the evolution of small disturbances to this steady state, denoting the disturbance tracer abundance, heating and streamfunction by $\tilde{\chi}$, $\tilde{Q}_s$, $\tilde{\psi}$, respectively. Linearising around the steady state gives,

$$\frac{\partial}{\partial t} \left( \frac{\partial^2 \tilde{\psi}}{\partial x^2} + \frac{f^2}{N^2} \frac{\partial^2 \tilde{\psi}}{\partial z^2} \right) = \frac{f^2}{N^2} \frac{\partial \tilde{Q}_s}{\partial z} - \frac{\alpha f^2}{N^2} \frac{\partial^2 \tilde{\psi}}{\partial z^2}, \tag{13a}$$

$$\frac{\partial}{\partial t} \left( \frac{\partial \tilde{\psi}}{\partial z} \right) + \frac{N^2}{f} \tilde{w} = Q_s - \alpha \frac{\partial \tilde{\psi}}{\partial z}, \tag{13b}$$

$$\frac{\partial \tilde{Q}_s}{\partial t} + \tilde{w} \frac{dQ_{s0}}{dz} = 0, \tag{13c}$$

where the final equation has substituted in for $\tilde{Q}_s = \tilde{\chi} \, d_z Q_{s0} / d_z \chi_0$. Note that having made the quasigeostrophic approximation, it is only the final equation that has been further linearised. Now, defining $\tilde{h} = \mathbb{R}\left( \hat{h}(z) \exp(\sigma t + ikx) \right)$, where $h \in (\psi, \chi, w)$, and substituting into (13), we obtain

$$\sigma \left( -k^2 \hat{\psi} + \frac{f^2}{N^2} \frac{d^2 \hat{\psi}}{dz^2} \right) = \frac{f^2}{N^2} \frac{d\hat{Q}_s}{dz} - \frac{\alpha f^2}{N^2} \frac{d^2 \hat{\psi}}{dz^2}, \tag{14a}$$

$$\sigma \frac{d\hat{\psi}}{dz} + \frac{N^2}{f} \hat{w} = \hat{Q}_s - \alpha \frac{d\hat{\psi}}{dz}, \tag{14b}$$

$$\sigma \hat{Q}_s + \hat{w} \frac{dQ_{s0}}{dz} = 0. \tag{14c}$$

Eliminating $\hat{w}$ from (14b) and (14c), substituting the resulting expression for $\hat{Q}_s$ into (14a) and defining $d_z S_0 = (f/N^2) d_z Q_{s0}$, we obtain

$$\frac{d}{dz} \left( \frac{\sigma + \alpha}{\sigma + d_z S_0} \frac{d\hat{\psi}}{dz} \right) = \frac{N^2 k^2}{f^2} \hat{\psi}. \tag{15}$$

This defines an eigenvalue problem for the growth rate $\sigma$. Intuitively, the eigenfunctions give the vertical structure of the vertical velocity as $\hat{w} = -(\sigma f/N^2)(\sigma + \alpha)/(\sigma + d_z S_0) d\hat{\psi}/dz$, i.e. equal to the expression in brackets on the left-hand side of (15) multiplied by $-\sigma f/N^2$.

First consider the case where there is no aerosol, i.e. $d_z S_0 = 0$. This system has been previously studied by Haynes and Ward (1993). Briefly, bounded solutions of (15) are plane waves. For a disturbance with vertical wavenumber $m$, (15) reduces to $\sigma = -\alpha/(1 + N^2 k^2/f^2 m^2)$ and hence $-\alpha < \sigma < 0$, meaning that the configuration is stable. Shallow disturbances, where $fm/Nk \gg 1$, have $\sigma \simeq -\alpha$ and hence decay at the radiative damping rate. In this limit the vertical velocity is small and the

dominant balance in (14b) is between the first term on the LHS and the thermal damping term. Deep disturbances, where $fm/Nk \ll 1$, have $0 < -\sigma \ll \alpha$ and hence are weakly decaying. Here, in (14a) the first term in the brackets on the right-hand side is much smaller than the second term, i.e. the contribution to the potential vorticity from the vertical temperature gradient is very small compared to the contribution from relative vorticity. Hence, the overall decay rate due to thermal damping is very small.

Next, consider the case where the heating has a constant vertical gradient, i.e. $d_z S_0 = -\gamma$. For a disturbance with vertical wavenumber $m$, (15) reduces to $\sigma = (\gamma N^2 k^2 - \alpha f^2 m^2)/(f^2 m^2 + N^2 k^2)$, meaning that $-\alpha < \sigma < \gamma$. There is instability if $\gamma > 0$; the fastest growth would be for deep disturbances where $|Nk/fm| \gg 1$. Physically, when $\gamma > 0$, the aerosol concentration decreases with height, i.e. $d_z S_0 < 0$. The instability arises because upward velocity at a given level brings air with greater aerosol abundances to that level, implying stronger heating and hence strong upward velocity: this effect is self-reinforcing.

There is instability if this effect is larger than that of thermal damping; whether this is true is determined by the ratio $|Nk/fm|$.

The scenario of constant $d_z S_0$ is not relevant to plausible atmospheric configurations where smoke generated by wildfires, or correspondingly aerosol generated by volcanic eruptions, is likely to be confined to layers of finite thickness, implying that regions of non-zero $d_z S_0$ will not only be confined in the vertical but, furthermore, will change sign, with positive values in the lower part of such layers and negative values in the upper part. Therefore (15) must be solved taking account of the non-trivial

form of $S_0(z)$, e.g., as a function that is positive in a localised region with a single maximum.

The problem is simplified a little by rewriting (15) in terms of the vertical velocity $\hat{w}(z) = -(f\sigma/N^2)(\sigma + \alpha)d_z\hat{\psi}/(\sigma + d_z S_0)$, giving

$$\frac{d^2 \hat{w}}{dz^2} = \frac{N^2 k^2}{f^2} \frac{\sigma + d_z S_0}{\sigma + \alpha} \hat{w}. \tag{16}$$

This is closely related mathematically to a standard 'potential well' problem in quantum mechanics, where the time-independent
Schrödinger equation is solved for a given form of the potential energy function to deduce the energy eigenvalues and corresponding wavefunctions. Here, given the form of $S_0$, solution of (16) implies the possible values of the eigenvalue $\sigma$ with the corresponding eigenfunctions describing the shape of the vertical velocity. It is straightforward to show that $\sigma$ is real. Solutions of interest may be oscillatory in $z$ away from the region of non-zero $S_0$, corresponding to 'unbound states', or they may decay away from the region of non-zero $S_0$, corresponding to 'bound states'. The latter may correspond to $\sigma > 0$ if $\sigma > -d_z S_0$
for some $z$. (16) may be solved numerically by multiplying by $\sigma + \alpha$, writing the second derivatives in finite-difference form and then solving the resulting standard-form matrix eigenvalue problem for $\sigma$. The maximum value of $\sigma$ corresponds to an eigenfunction $\hat{w}$ which has no zeros, as is standard in such eigenvalue problems, implying that the most unstable mode has at each $z$ either ascent or descent for all $x$. For a specified $S_0(z)$, varying on a length-scale $D$, there is always at least one bound

state, corresponding to $\sigma > 0$. Asymptotic analysis set out in the Appendix, in which $\hat{w}$ is expanded in a small parameter, $(NkD/f)^2$, and solved with matching conditions, shows that in the limit as $NkD/f \to 0$ there is a single bound state and the corresponding expression for $\sigma$ is

$$\sigma \simeq \frac{1}{4}\alpha \left(\frac{Nk}{f}\right)^6 \left(\int_{-\infty}^{\infty} S_0^2 \, dz\right)^2. \tag{17}$$

As $NkD/f$ increases, more and more bound states emerge and the maximum $\sigma$ increases towards a constant value equal to $-\min\{\mathrm{d}_z S_0\}$. The difference from its maximum value varies as $k^{-1}$ as described by the analytical approximation for $\sigma$ in this limit,

$$\sigma \simeq -\mathrm{d}_z S_{0m} - (\alpha - \mathrm{d}_z S_{0m})^{1/2} \frac{f}{Nk\sqrt{2}} (\mathrm{d}_z^3 S_{0m})^{1/2}. \tag{18}$$

Details of the derivation are in the Appendix.

The behaviour can be illustrated by considering a Gaussian vertical profile, $S_0(z) = D\alpha e^{-z^2/D^2}$. The maximum growth rate $\sigma \ (= -\min\{\mathrm{d}_z S_0\})$ is then equal to $\alpha$ multiplied by a function of $NkD/f$. The equation (16) is solved with vanishing boundary conditions, where $\hat{w} = 0$ at large finite negative and positive values of $z$, approximating $z \to \pm\infty$. When $NkD/f$ is small, the spatial decay of the solution away from $z = 0$ is very slow and the boundary values of $z$ must therefore be taken to be very large. Figure 10($a$) shows the nonlinear growth rate as a function of $NkD/f$ in this case, with the asymptotic expressions (17) and (18), for small and large $NkD/f$ respectively, superimposed as well as the maximum growth rate. Note the main qualitative features: the growth rate is very small for small $NkD/f$ and it tends to a finite maximum value as $NkD/f \to \infty$. Inclusion of a scale-dependent dissipation, for example, diffusion acting on the tracer field, will lead to a maximum growth rate at a finite value of $NkD/f$.

Figure 10($b$) shows the form of the eigenfunction $\hat{w}$, for selected values of $NkD/f$. The eigenfunctions are centred on the region where $\mathrm{d}_z S_0$ is negative such that positive vertical velocity peaks there, which is consistent with negative values of $\mathrm{d}_z S_0$ leading to instability. When $NkD/f$ is small, the non-zero vertical velocity anomaly extends a large distance from the region of negative $\mathrm{d}_z S_0$, consistent with the fact that growth rates are weak. As $NkD/f$ increases, the vertical velocity anomaly becomes increasingly confined to the region of negative $\mathrm{d}_z S_0$ and is localised about its minimum value. This narrowing of the vertical velocity peak is consistent with the fact that the growth rate in this limit tends to $-\min\{\mathrm{d}_z S_0\}$ (matching the results noted above for the case in which $\mathrm{d}_z S_0$ is constant in $z$).

The structures simulated here have some resemblance to those seen in simulations of the maintenance of tropopause-level cirrus clouds by the circulation, forced by cloud radiative heating via the interaction of cloud dynamics, microphysics, and

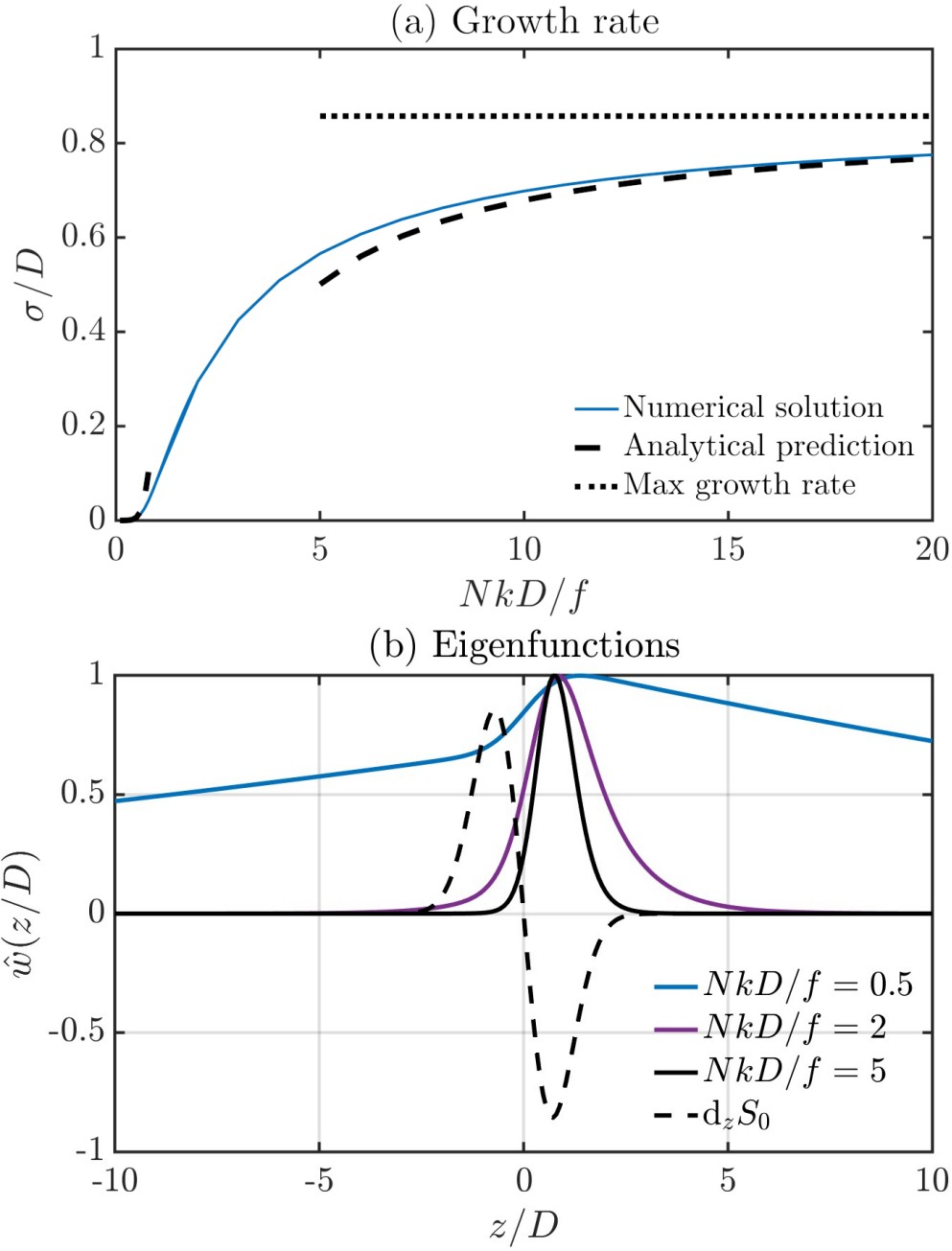

**Figure 10.** (a) Growth rate as a function of $NkD/f$ for $S_0(z) = \alpha D e^{-z^2/D^2}$. Numerical evaluation by solving the discretized eigenvalue problem is the solid line. Dashed lines for small and large $NkD/f$ show analytical expressions according to (17) and (18) respectively. The dotted line is the maximum growth rate, $-\min\{d_z S_0\}$. (b) Solid curves plot eigenfunctions $\hat{w}(z)$, giving the shape of vertical velocity for $NkD/f = 0.5$, 2 and 5, for the Gaussian $S_0(z)$. The dashed curve is $d_z S_0$. Note that only part of the complete $z$-domain, which is $[-40D, 40D]$, is shown. The amplitude of $\hat{w}$ is arbitrary and, for display, has been chosen such that the maximum value is 1. The shapes of the eigenfunctions become narrower as $NkD/f$ increases and increasingly localised near the minimum of $d_z S_0(z)$.

radiation (e.g., Dinh et al., 2010). However the mechanisms operating are very different. There the scales are much smaller, there is active convection, and there is no role for rotation. In our case rotation plays a key role in the dynamics.

## 4.2 Finite amplitude disturbances

Having established that there is unstable growth of small disturbances to a horizontally homogeneous tracer layer, the be-
715 haviour when the disturbances reach finite amplitude may be investigated by solving the complete quasigeostrophic system (13), without the linearisation assumption in the Cartesian form of the tracer equation (2). Figure 11 shows the evolution of an aerosol layer, initialised with Gaussian form in the vertical, with a superimposed disturbance, at 5, 10, 30 and 50 days. The superimposed disturbance is obtained by multiplying the Gaussian profile value at each grid point by an independent random number drawn from a uniform distribution with range $[0.5 : 1.5]$.

The early stages of the evolution seen in Figure 11 demonstrate the growth expected from the linear stability analysis, given that the growth of large wavenumbers is expected to be inhibited by tracer diffusion effects. What is seen subsequently is the ascent of isolated plumes of tracer out of the location of the initial layer. The lower part of the layer remains relatively undisturbed, consistent with the expectation that the increase in tracer concentration with height in that part of the layer is stabilizing rather than destabilizing. As time increases further, the distance of penetration of the plumes increases. There is
also indication that the horizontal scale of the plumes increases with time, suggesting a self-organisation of the flow. The tracer plumes would be wider if the initially horizontal layer were thicker.

These two-dimensional results notwithstanding, as we have previously noted, the assumption of two-dimensionality, whether in Cartesian geometry or as axisymmetry, misses potentially important mechanisms such as vortex isolation, aerosol bubbles pinching off ascending plumes, coalescence of plumes. A more complete study would require analysis of the full 3-D problem.

## 5 Discussion and conclusions

Following penetration of wildfire smoke or of volcanic aerosol into the stratosphere, recent studies have detected evidence of aerosol-filled anticyclonic vortices that persist for several weeks and ascend for large distances, typically 10-20km. Aerosols are known to be effective absorbers of radiation and their presence in large concentrations will therefore give substantial heating effects at the location of these aerosol-filled vortices. Various important details of the observed dynamical structures require
further explanation, such as the fact that a single-signed anticyclonic potential vorticity anomaly is co-located with a localised heating and the ascent of the vortex across isentropic surfaces, which cannot be explained by material conservation of potential vorticity.

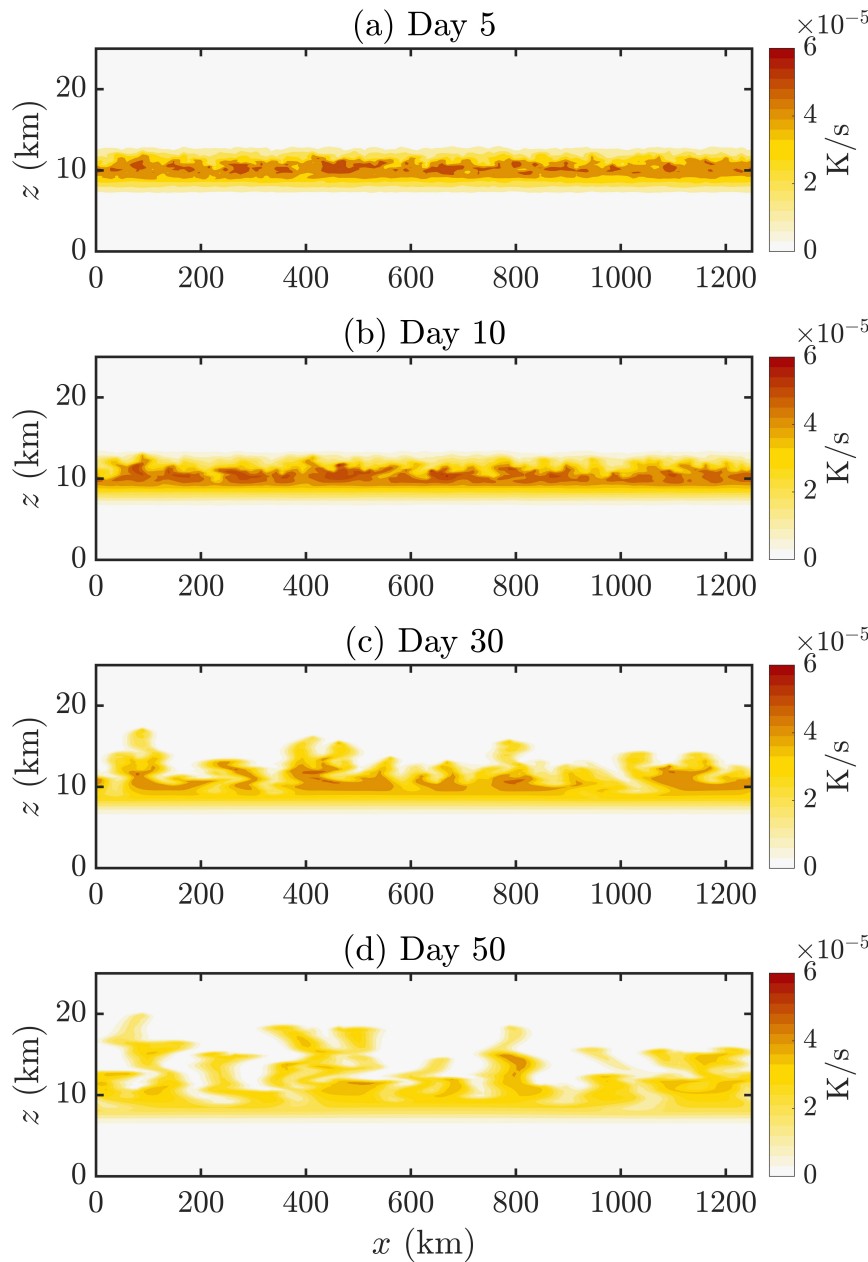

**Figure 11.** Solutions of the full quasigeostrophic equations (13) in Cartesian coordinates with periodic boundary conditions, showing the tracer distribution at (a) 5, (b) 10, (c) 30, (d) 50 days, as it evolves from an initial condition of a horizontally homogeneous layer disturbed by superimposed random noise. Tracer diffusivity values are $\kappa_h=0.5\times10^2\,\mathrm{m}^2\mathrm{s}^{-1}$; $\kappa_v=0.5\times10^{-1}\,\mathrm{m}^2\mathrm{s}^{-1}$.

In this paper we have considered a simplified dynamical description of these vortices, starting with an assumption of axisymmetry together with hydrostatic and gradient wind balance, which leads to the classical Eliassen problem for the response of a vortex in a rotating stratified fluid to applied heating. The novel ingredient here is that the heating is determined by a tracer, representing sunlight-absorbing aerosol, which is transported upward by the secondary circulation, which is itself part of the Eliassen response to the tracer heating. There is therefore a two-way coupling between the evolution of the tracer and the circulation. In reality the observed aerosol-driven vortices are contained within a larger scale three-dimensional stratospheric flow, which is likely to have a strong deforming effect. Hence, the assumption of axisymmetry has several limitations which are discussed in more detail below. In particular, an axisymmetric theory cannot account for the isolation of tracers within strong vortices, which is a well-known phenomenon in geophysical fluid dynamics and is likely to be a major part of an overall description. A further simplification in most of our explicit calculations is that we use the QG form of the Eliassen problem rather than the full non-QG form. This may limit the applicability of some of the detailed predictions, though not necessarily the broader quantitative predictions such as scaling estimates.

In §1 we highlighted four key specific questions which motivate further study of the dynamics of aerosol-filled vortices: (i) How does an isolated anticyclonic vortex emerge as a response to heating and why is the anticyclonic vortex apparently centred at the same level as the heating rather than above it? (ii) What determines the rate of rise of the tracer anomaly and accompanying anticyclonic vortex? (iii) What determines the strength of the vortex and the corresponding temperature anomaly? (iv) Once aerosol is injected into the stratosphere, what is the mechanism for its organisation into long-lived ascending heating-driven vortex structures and under what conditions is this organisation likely to take place? We now proceed to address each question in turn, highlighting insights from our dynamical formulation, what our formulation does not capture, and avenues for future research.

Regarding question (i), the axisymmetric model provides a clear explanation. An upward moving localised heating field provides an upward moving dipolar potential vorticity forcing, anticyclonic above and cyclonic below. The effect of the ascent is to give an anticyclonic potential vorticity anomaly moving upward with the heating and leaving behind a stationary cyclonic anomaly just below the initial location of the heating. The upward movement of the anticylonic PV anomaly is not a result of material conservation of potential vorticity but results instead from the heating-induced forcing. We demonstrated this first by omitting the tracer-dynamical coupling and simply specifying upward motion of the heating field, both as the solution to an initial value problem and as an analytical steadily translating solution of the dynamical equations. We then included the tracer-dynamical coupling and showed the evolution of an initial tracer, and hence heating field, led to ascent of the top part of the tracer distribution with an accompanying anticyclonic vortex.

With respect to question (ii), for our self-lofting scaling estimates to hold, a minimum heating rate $Q_s \gg N^2 H_0/R \times \alpha D \times f^2 L^2/N^2 D^2$ is required. When self-lofting, the rate of ascent of the tracer-filled vortex is proportional to the magnitude of the tracer-associated heating, bearing in mind from (4e) that only part of the heating is balanced by upwelling. In the coupled tracer-dynamics model simulations presented here, the assumed tracer concentration corresponds to maximum heating rates of $5 \times 10^{-5} \text{Ks}^{-1}$. If this was balanced by upwelling, the magnitude of the latter would be about $5 \times 10^{-3} \text{ms}^{-1}$. The ascent actually observed is about half of this value, consistent with the fact that the ascending tracer anomaly, and hence heating, has a ratio of vertical to horizontal scales that is roughly equal to $f/N$. (7) predicts that the vertical velocity scales as $RQ_s/N^2 H_0$, broadly consistent with the observed vertical velocities during the early and later stages of the main vortex evolution originating from the Australian wildfires. In short, our dynamically consistent model has verified that the ascent rate is generally in agreement with estimated heating rates (as suggested by others, e.g., Khaykin et al., 2020; Lestrelin et al., 2021).

For question (iii), the scaling analysis presented in this study suggested that the quasigeostrophic potential vorticity magnitude (and relative vorticity magnitude) in the tracer-filled vortex would be $O(f)$. (This estimate corresponds, for an anticyclone, to near-zero full PV (Berrisford et al., 1993).) This estimate is consistent with observations, e.g., at $45^o$ S our prediction gives $\zeta \sim O(10^{-4})\ \text{s}^{-1}$, matching the magnitude of peak vorticity of the main vortex associated with the Australian wildfires (e.g., Figure 6 in Khaykin et al. (2020)) and the suggestion of zero full PV in Lestrelin et al. (2021). This estimate was also consistent with our model simulations, albeit subject to the caution that most of the simulations presented assumed the quasigeostrophic approximation, which breaks down for vorticity of this magnitude. However the limited-duration non-quasigeostrophic simulations presented in §2.1 showed vorticity values approaching $f$ prior to the breakdown of the balanced equations, manifested by the potential vorticity values in the anticyclone approaching zero. The amplitude of other dynamical measures follows from the magnitude of vorticity being $O(f)$ and, in particular, are independent of the heating magnitude $Q_s$. Azimuthal velocities scale as $fL$, where $L$ is the horizontal scale of the vortex. Temperatures scale as $(H_0 f/R) fL^2/D = f N L H_0/R$, with $D$ being the vertical scale of the vortex, and the equality following from assumption of $f/N$ as the ratio of vertical scales to horizontal scales. For $v$ and $T$, estimates from these scalings and typical values of $v$ and $T$ in our time integrations are in agreement with composites in Figure 6(df) of Khaykin et al. (2020). For the parameter values chosen, corresponding to maximum heating rates of order $10^{-5} \text{Ks}^{-1}$ and ascent rates of order $10^{-3} \text{ms}^{-1}$, and tracer anomalies with vertical scale of a few km, strong thermal damping was found to increase the ascent rate of the leading edge of the tracer structure (as pointed out by Podglajen et al. (2024)) and, of course, some of the details of the temperature structure were altered. On relaxing the Boussinesq approximation and taking account of density variation with height, which led to bubble expansion and less diffusive reduction of tracer, high tracer concentrations (i.e. heating rates) were maintained for longer time, leading not only to stronger ascent but also to corre-

spondingly deeper vorticity and temperature anomalies. What the axisymmetric model did not demonstrate was a convincing confinement of the tracer to the interior of the anticyclonic vortex. In the examples shown in Figure 5, an ascending tracer-filled vortex emerged out of the initial tracer distribution, but it did not detach clearly from the larger tracer distribution which was also transported by the secondary circulation. (A deeper initial distribution may do better in this respect.) We suggest, as have others, that in the real atmosphere the stirring and mixing outside the tracer-filled vortex plays an important role. As a crude ad-hoc representation of this in the axisymmetric model, we simply removed any tracer outside of the vortex as defined by a specified threshold value of potential vorticity, describing this as a vortex-stripping adjustment. The persistence and ascent of the tracer-filled vortex was robust to this adjustment.

Turning now to question (iv), as noted above, the axisymmetric model had some success in demonstrating the emergence of a detached, ascending tracer-filled vortex for suitable conditions on the initial tracer distribution. So, it could be the case that the details of injection (which we do not address), which sets up the initial tracer distribution, is key to the emergence of the tracer-filled vortices. We also investigated another possibility: that the coupling between tracer and heating played an active role in the initial stages of vortex emergence. We considered the stability of an initially horizontal layer of tracer, which is arguably the most general initial profile that can be considered. The configuration was shown to be unstable, as a result of self-reinforcement between heating and ascent at levels where the tracer concentration was decreasing with height. Furthermore, as the disturbances reached finite-amplitude, they resulted in the break-up of the tracer layer and the formation of rising structures of tracer plumes. Over time, these plumes penetrated increasingly deep into the stratosphere and their horizontal lengthscale appeared to increase. The model in this case was two-dimensional which, as for the axisymmetric model, implied absence of any vortex isolation or vortex stripping effects. Inclusion of such effects would require a three-dimensional calculation. Nonetheless, what seems likely from our results is that tracer-filled vortices emerge as a result of a combination of various effects, which include the two-way interaction between tracer and dynamics as demonstrated in the axisymmetric and two-dimensional models, the geometry of the tracer injection into the stratosphere, and the effects of the background stratospheric flow.

As has been noted previously, evidence from studies of vortex isolation and vortex stripping suggest that a vortex is more likely to remain coherent and to isolate tracer within it if the vorticity magnitude is sufficiently large relative to external shear and strain rates. The conclusions above are that the typical relative vorticity of an ascending anticyclonic vortex, once it has formed, is $O(f)$, i.e. $O(10^{-4})$ s$^{-1}$, for typical midlatitude parameters, and independent of the heating rate associated with the tracer. The lack of dependence of the typical vorticity magnitude on the heating rate follows from two findings: that both the vorticity (or potential vorticity) forcing and the timescale on which this forcing is experienced at a given level are proportional

to the heating rate, and that the vorticity magnitude in the steady-state ascending phase depends on the ratio of these two quantities. What does depend on the heating rate is the time taken for the vorticity to reach $O(f)$, estimated previously in §3.2 as $DH_0N^2/Q_sR$, i.e. inversely proportional to the heating rate. The implication is that, once ascending tracer-filled vortices have formed, the likelihood of them remaining coherent and isolated within the background stratospheric flow is independent of the heating rate due to the tracer. However because the time taken for formation will be longer when the heating rate is smaller, tracers corresponding to larger heating rates are more likely to reach the coherent ascending stage while tracer anomalies with weak heating rates are more likely to be pulled apart by the large-scale flow and mixed into the background environment during the early formation stage.

As we have noted previously, as we submitted this paper for publication, we became aware of an independent study on the dynamics of diabatically-driven stratospheric anticyclones that had been submitted elsewhere (Podglajen et al., 2024) and in revising our paper we have taken account of that study, which combines numerical simulation in a 3-D non-hydrostatic model with a simple theoretical model based on an elegant mathematical approach. In §3.6, we gave a brief summary of the principal similarities and differences in methods and conclusions between the two studies. The differences in choice of initial condition between P2024 and our study were noted and we return to that here. P2024 argue that the injection of tropospheric air into the stratosphere as a homogeneous intrusion (Gill, 1981) will by itself give an anticyclonic PV anomaly, without any effect of tracer heating. They correspondingly initialise with a low PV anomaly. While this is a valid point, whether or not this is the only appropriate initial condition is still open to question. For one thing, radiative transfer, perhaps enhanced by the effects of vertical shear (Haynes and Ward, 1993), might act to dissipate the initial PV anomaly, but an aerosol cloud would remain and its heating effects then force an upward moving anticyclone. Correspondingly for highly non-axisymmetric evolution, such as that arising from several aerosol plumes and initial vortices combining to form a single aerosol cloud, as envisaged by P2024, it might be that the initial vortices would be dissipated but the forcing effect of the aerosol cloud, even if it were dispersed over a significantly larger volume than the original injections, would remain and be sufficient to form and sustain an upward propagating anticyclone. In our study, the aerosol-filled vortex problem we considered in §3 and the aerosol layer problem we considered in §4 have relevance to the self-organisation of aerosol clouds in the stratosphere.

The models presented in this paper are intentionally simplified and we conclude by discussing how these simplifications might be relaxed. One key choice was that the heating was proportional to the tracer concentration. This was based on the assumption that the heating arises primarily from short-wave absorption by wildfire aerosols, such as black carbon, sulphate aerosols, etc. Calculations by Sellitto et al. (2023), for example, provide much more detail on this heating by taking account of the optical properties of the aerosol layer (which in turn depends on its composition, e.g., quantities of black carbon, brown

carbon). Alongside this, we represented the effects of long-wave radiative transfer here by a constant thermal damping rate. Of

855 course, the details are more complicated, and the constant damping rate might be replaced by, for example, a scale-dependent damping rate (e.g. Haynes and Ward, 1993). In our study, parameter values were chosen to be relevant to the observations of long-lived anticyclonic vortices driven by strong heating; the role of thermal damping for vortices with weak heating effects remains to be explored (e.g., the range of Rossby numbers and vortex strengths for the Canadian wildfire's vortices in Table 1 of Lestrelin et al. (2021) suggest a variety of additional problems to consider).

As has been noted at various points in the paper, a simplification made here is the neglect of 3-D effects, principally the competing effects of vortex isolation and the distortion of vortices by external strain and shear, leading to stripping away of the outer layers of vortices and eventually to vortex destruction. For example, the role of vertical shear has been noted in breaking apart (Khaykin et al., 2020) and in elongating (Lestrelin et al., 2021) tracer features assumed associated with aerosol-filled vortices, as well as discussed in the context of initial tilting and subsequent evolution of an anticyclonic vortex (Allen et al.,

2020). These processes may be important in the early stages of vortex formation when the magnitude of vorticity is growing, as the likelihood of its survival is linked to this initial rate of vorticity increase. Precisely reproducing observed evolution will be difficult to model deterministically due to the sensitive dependencies on the flow details, which themselves will vary strongly from one event to another. Nonetheless in the case of recently reported numerical simulations such as those in Doglioni et al. (2022), it would be valuable to see more detail of the vortex formation processes in the simulation, regardless of whether they

match what occurred in the real atmosphere. Finally it would be very valuable to have results from 3-D simulations of the type in Podglajen et al. (2024) to gain insight into the physical processes driving the internal structure of the heating-driven vortices.

## Appendix A: Analysis of equation (16)

The equation (16) may be rewritten as

$$-\frac{d^2\hat{w}}{dz^2} + \frac{N^2 k^2}{f^2}\frac{d_z S_0}{\sigma + \alpha}\hat{w} = -\frac{N^2 k^2}{f^2}\frac{\sigma}{\sigma + \alpha}\hat{w} \tag{A1}$$

and recognised as a Schrodinger equation with potential $N^2 k^2 d_z S_0 / f^2(\sigma + \alpha)$ and energy eigenvalue $-N^2 k^2 \sigma / f^2(\sigma + \alpha) = -\lambda^2$. There are two natural limiting cases, one when the scale $\lambda^{-1}$ is much larger than the scale, $D$ say, on which $S_0$ varies and the other when $\lambda^{-1}$ is much smaller than $D$.

### A1   $NkD/f \ll 1$

In this case the variation of $\hat{w}$ is weak in the region $z \sim D$. Defining $l = z/D$, a natural approximation sequence is $\hat{w} = \hat{w}_0(l) + (NkD/f)^2 \hat{w}_1(l) + (NkD/f)^4 \hat{w}_2(l) + \ldots$. Proceeding order-by-order, we hence obtain,

$$-\frac{d^2\hat{w}_0}{dl^2} = 0, \tag{A2a}$$

$$-\frac{d^2\hat{w}_1}{dl^2} + \frac{d_z S_0}{\sigma + \alpha}\hat{w}_0 = 0, \tag{A2b}$$

$$-\frac{d^2\hat{w}_2}{dl^2} + \frac{d_z S_0}{\sigma + \alpha}\hat{w}_1 = 0. \tag{A2c}$$

In $|z| \gg D$, the leading order approximation is $-d^2\hat{w}/dz^2 = -\lambda^2\hat{w}$, hence $\hat{w} \simeq A_- e^{-\lambda z}$ in $z < 0$ and $\hat{w} \simeq A_+ e^{\lambda z}$ in $z > 0$. The solution in $z \sim D$, provided by (A2a), needs to be matched to that in $|z| \gg D$, which requires the matching condition $A_- = A_+ = \hat{w}_0$ (with $\hat{w}_0$ a constant). Hence it is required that $[d_z\hat{w}]_{0-}^{0+} = -2\lambda\hat{w}_0$. Integrating (A2b) and matching implies that $[d_z\hat{w}]_{0-}^{0+} = (Nk/f)^2 D[d_l\hat{w}_1]_{-\infty}^{\infty} = (\int_{-\infty}^{\infty} d_z S_0)\,dz)(N^2 k^2/(Df^2(\sigma + \alpha)))\hat{w}_0 = 0$, since $S_0(z) \to 0$ as $z \to \pm\infty$. Hence the leading order contribution to $[d_z\hat{w}]_{0-}^{0+}$ is determined by $\hat{w}_2$. Integrating (A2c) by parts, and then substituting for $d_z\hat{w}_1$ from the integral of (A2c) gives

$$\left[\frac{d\hat{w}}{dz}\right]_{0-}^{0+} = \frac{N^2 k^2}{f^2(\sigma + \alpha)}\int_{-\infty}^{\infty} d_z S_0 \hat{w}_1\,dz = -\frac{N^4 k^4}{f^4(\sigma + \alpha)^2}\hat{w}_0\int_{-\infty}^{\infty} S_0^2\,dz, \tag{A3}$$

where integrals in $l$ have been re-written in terms of integrals in $z$. This implies that

$$\frac{N^4 k^4}{f^4(\sigma + \alpha)^2}\int_{-\infty}^{\infty} S_0^2\,dz = \frac{2Nk\sigma^{1/2}}{f(\sigma + \alpha)^{1/2}} \tag{A4}$$

and hence that

$$\sigma \simeq \frac{1}{4}\alpha\left(\frac{Nk}{f}\right)^6\left(\int_{-\infty}^{\infty} S_0^2\,dz\right)^2. \tag{A5}$$

 **A2** $NkD/f \gg 1$

In this case the vertical velocity $\hat{w}$ varies strongly in the region $z \sim D$. The eigenfunction corresponding to the largest growth

rate is localised near the minimum of $\mathrm{d}_z S_0$. Assume that this is located at $z = z_{0m}$, where $\mathrm{d}_z S_0 = \mathrm{d}_z S_{0m}$ and $\mathrm{d}_z^3 S_0 = \mathrm{d}_z^3 S_{0m}$.

Then $(N^2 k^2 / f^2)\, \mathrm{d}_z S_0/(\sigma + \alpha)$ may be expanded in a Taylor series about $z = z_{0m}$, to give

$$\frac{\mathrm{d}^2 \hat{w}}{\mathrm{d}z^2} = \frac{\sigma + \mathrm{d}_z S_{0m}}{\sigma + \alpha}\frac{N^2 k^2}{f^2}\hat{w} + \frac{\mathrm{d}_z^3 S_{0m}}{\sigma + \alpha}\frac{N^2 k^2}{2f^2}(z - z_{0m})^2 \hat{w} + \dots$$

$$= E_0(\sigma)\hat{w} \qquad\qquad + E_2(\sigma)(z - z_{0m})^2 \hat{w} \qquad + \dots, \tag{A6}$$

with the second equality defining the expressions $E_0$ and $E_2$.

Rewriting (A6) in terms of the $\xi = (E_2)^{1/4}(z - z_{0m})$ gives at leading order

$$\frac{\mathrm{d}^2 \hat{w}}{\mathrm{d}\xi^2} - \xi^2 \hat{w} = (E_2)^{-1/2} E_0 \hat{w} \tag{A7}$$

which is a Hermite equation with bounded solutions only when $(E_2)^{-1/2} E_0 = -(2n+1)$ where $n = 0, 1, 2, \dots$. The choice

$n = 0$ corresponds to the eigenfunction with no zeros and hence to the eigenvalue with largest growth rate. This relation

between $E_0$ and $E_2$ implies that

$$\frac{\sigma + \mathrm{d}_z S_{0m}}{\sigma + \alpha}\frac{N^2 k^2}{f^2} = -\left(\frac{\mathrm{d}_z^3 S_{0m}}{\sigma + \alpha}\frac{N^2 k^2}{2f^2}\right)^{1/2}. \tag{A8}$$

Then using the leading-order approximation that $\sigma = -\mathrm{d}_z S_{0m}$ it follows that an improved approximation for $NkD/f \gg 1$ is

$$\sigma \simeq -\mathrm{d}_z S_{0m} - (\alpha - \mathrm{d}_z S_{0m})^{1/2}\frac{f}{Nk\sqrt{2}}(\mathrm{d}_z^3 S_{0m})^{1/2}. \tag{A9}$$

This expression shows how the growth rate approaches the maximum value as $NkD/f$ increases.

*Author contributions.* K.S.S. and P.H.H. designed research, performed research, analysed solutions, and wrote the paper.

*Competing interests.* The authors declare that they have no conflict of interest.

*Acknowledgements.* K.S.S. gratefully acknowledges a postdoctoral fellowship from the James S. McDonnell Foundation. The authors thank
one anonymous reviewer and Bernard Legras for reviewing this study, and Thomas Birner for serving as editor.

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
