# Peer review of "How heating tracers drive self-lofting long-lived stratospheric anticyclones: simple dynamical models"

_EGUsphere, 2023_

## Referee Comment (RC1)

Review of "How heating tracers drive self-lofting long-lived stratospheric anticyclones: simple dynamical models," by Kasturi S. Shah and Peter H. Haynes.

**General Comments**

This article examines the dynamical response to heating induced by solar absorption of wildfire smoke or volcanic aerosols in the stratosphere. The motivation is to understand various observed aspects of newly discovered tracer-driven anticyclones. Using a simple axisymmetric model, the authors are able to reproduce many of the salient features of these phenomena. In particular, they provide an explanation for the observation of a single-signed potential vorticity anomaly and vertical temperature dipole. Another 2-D model is used to explore the initial stages of formation. The paper is well written and provides useful new insight into these newly-discovered phenomena.

**Specific Comments**

Line 38: You often say "smoke or aerosol". Isn't smoke considered an aerosol?

For the cooling term (Eq. 3), is the T relative to the background temperature so that if T=Tb, then you wouldn't get cooling?

Line 263: What do you mean by "weakly stable configuration".

Line 281: Is the heating actually in units of K/day? This seems like an extremely small heating rate. If the heating was K/s, that would translate to ~4 K/day, which is on par with the observed heating. I saw later in the discussion that you use the units of K/s (line 669), so I assume that is true throughout the paper. Please check units in text and figures.

Figure 2b: It is difficult to see the symbols for tracer maximum and QGPV minimum. Maybe in both 2a and 2b you could place these at ~600 km on the x-axis so they are easily seen?

Figure 2c:

- The grey line is very difficult to see.
- The initial anticyclone extends all the way up to 40 km, albeit small. Is this expected?
- The QGPV minimum appears to have a slight oscillation with period of a few days. Is this just a numerical artifact?
- In the limit of W small, but integrated a long time, do the two PV anomalies rise together, or do they separate as seen here?
- Might help to label the color bar with units.

Figure 3:

- Is the radial location of the minimum temperature always at r=0?
- Is this the absolute temperature minus the background temperature?
- Again, hard to see the grey line.
- May want to label the units either near the color bar or in the caption.
- Text refers to Figure 3a on line 305, but there is no 3a in the actual figure. May want to label all the panels in this figure and other figures to help the reader.

- On the right column, the labels say "Steadily translating solution v numerical solution". But this is q, not v, right?
- For the right column, are the units consistent with the labeled values on the contours?

Line 333-4: I'm confused how the estimate for the PV anomaly is f. Why doesn't this depend on the heating rate, since in the limit of zero heating wouldn't there be no PV anomaly. I see you discuss this further in the last section, but it still doesn't make intuitive sense.

Line 384: The diffusivities are constant, yet horizontal diffusivity in the stratosphere has been shown to vary, e.g., having minima associated with significant barriers to horizontal mixing associated with vortices (e.g., Haynes and Shuckburgh, 2000). Can the authors comment on whether you would expect similar diffusivity structures in these newly discovered tracer-driven vortices? Is there a way you could simulate this effect in your model by having diffusivity depend on model variables, such as PV gradient?

Figure 4:

- May want to include labels for (a), (b), …, (f) on the figure
- Are the units for heating correct, about 10^-5 K/day or should they be K/s?
- Hard to see tracer maximum and QGPV minimum symbols.
- The tracer distributions have what appears to be a very narrow maximum and then a broader elevated region, which is different from the observations. In the top and bottom cases there are actually two distinct maxima. Do you have any comment on this structure?
- It appears the temperature minimum is co-located with the QGPV minimum. Is that correct? If so, this is also different from observations.

It might be useful to plot a schematic diagram, similar to Figure 1b, for this simulation. This would help the reader to compare more directly with observations.

Is there a significant radial wind in these simulations? It would be nice to see the complete u, v, w structure.

The vortex stripping experiments are used to help remove tracer that occurs well below the maximum. How much of a role does vertical diffusion play in this remnant low-level tracer? What happens if you run the model with very small or very large vertical diffusion?

Figure 5:

- May want to include labels for (a), (b), (c), (d) on the figure
- On line 442 I think you have the panels reverse: (a) should be without, (b) should be with
- Does the temperature dipole structure have better agreement to observations in this case than in the previous? Is there cooling above and warming below the QGPV minimum? Again, a schematic like Figure 1b would be nice.

Figure 6:

- Again, may want labels on the individual figure panels.
- Again, it would be interesting to see the structures in the wind fields for this case.

- Again, the temperature dipole does not agree with Figure 1b. The minimum peaks at the same location as the PV minimum. Any idea why this is the case?

The discussion of the instability in the x-z plane was very interesting. I wonder if it would be beneficial (not for this paper, but for future work) to also perform the 2-D analysis in the x-y plane (e.g., shallow water system) where a heating tracer is initially placed in an irrotational flow. The heating could increase the layer thickness and cause rotation via geostrophic balance, which would help to contain the tracer against horizontal shearing flow. This wouldn't model the lofting effect but could help to understand some of the self-organizing flow.

Line 673: There were at least three vortices from the 2019/2020 Australian wildfires. I think you mean the largest vortex, since the others ascended at a slower rate and didn't go as high or last as long.

Lines 687-689: Might be useful to quote your typical azimuthal velocity and temperatures produced in your simulations and compare them to observations.

Lines 720-722: Some discussion of vortex resistance to shear was also presented in Allen et al. (2020), https://doi.org/10.1175/JAS-D-20-0131.1 which was a follow-up to Kablick et al. (2020) and provided more detailed analysis of dynamical properties of the main anticyclone associated with the Australian New Year event, in addition to examining one of the 2017 Pacific Northwest Event plumes.

**Technical Corrections**

Line 3: should be "carbon monoxide"

Line 30: should be "carbon monoxide"

Line 31: Should be "have shown" to match the subject "fields"

Line 148: "a r a" sounds funny. Maybe remove first "a"

Figure 1 caption: "and volcanic eruptions" can be removed before the first set of references

Line 330: remove "is" from "ascent is tends"

---

## Author Response (AR1)

**Response to Reviewer 1**

Review of "How heating tracers drive self-lofting long-lived stratospheric anticyclones: simple dynamical models," by Kasturi S. Shah and Peter H. Haynes.

**General Comments**

This article examines the dynamical response to heating induced by solar absorption of wildfire smoke or volcanic aerosols in the stratosphere. The motivation is to understand various observed aspects of newly discovered tracer-driven anticyclones. Using a simple axisymmetric model, the authors are able to reproduce many of the salient features of these phenomena. In particular, they provide an explanation for the observation of a single-signed potential vorticity anomaly and vertical temperature dipole. Another 2-D model is used to explore the initial stages of formation. The paper is well written and provides useful new insight into these newly-discovered phenomena.

Thank you for your thorough review of our manuscript. We have provided a point-by-point response to your review below, accompanied by Figures R1 and R2. All line numbers in this response file refer to the *revised* manuscript, unless otherwise indicated. We have also uploaded our revised manuscript as well as a marked-up version of the revised manuscript in which additions to the text are marked in deep green colour.

**Specific Comments**

Line 38: You often say "smoke or aerosol". Isn't smoke considered an aerosol?

We agree. We have now switched every instance of "smoke or aerosol" and "smoke" to "aerosol" in the main text. On L40 we say "...volcanic aerosol and wildfire smoke (henceforth collectively referred to as 'aerosol')...".

For the cooling term (Eq. 3), is the $T$ relative to the background temperature so that if $T = T_b$, then you wouldn't get cooling?

$T$ is the temperature anomaly relative to the background state (see the definition of $T$ on L173-174).

Line 263: What do you mean by "weakly stable configuration".

We have now edited L312 for clarity (added text italicised): "The observed temperature dipole has a cold anomaly above the heating and warm anomaly below, *implying reduced static stability.*"

Line 281: Is the heating actually in units of K/day? This seems like an extremely small heating rate. If the heating was K/s, that would translate to 4 K/day, which is on par with the observed heating. I saw later in the discussion that you use the units of K/s (line 669), so I assume that is true throughout the paper. Please check units in text and figures.

Thank you for spotting this typo. We have corrected units for the heating rate in the figures and in the text.

Figure 2b: It is difficult to see the symbols for tracer maximum and QGPV minimum. Maybe in both 2a and 2b you could place these at $\approx 600$ km on the $x$-axis so they are easily seen?

We agree that the symbols were hard to see, and have now removed these symbols from all relevant figures of the manuscript as the region of tracer maximum and QGPV minimum is clearly visible by eye.

Figure 2c: (Note this is now Figure 3($c$) in the revised manuscript.)

- The grey line is very difficult to see.

  Done: the tracer centroid is now plotted in a solid black line and the anticyclone centroid in a solid blue line in all figures for improved readability.

- The initial anticyclone extends all the way up to 40 km, albeit small. Is this expected?

  This was an artefact of the large contour interval used in this figure. There is near-zero anticyclonic PV extending up to 40 km. We have now reduced the size of the contour intervals in all figures.

- The QGPV minimum appears to have a slight oscillation with period of a few days. Is this just a numerical artifact?

  The zig-zaggedness is a numerical artifact associated with the finite resolution of the $(r, z)$ grid on which the equations are solved. The level of the minimum must be at one of the grid points, therefore with a feature moving upwards very slowly the position will inevitably make discrete jumps. Using higher resolution shows that there is no significant inaccuracy in the representation of the solution, e.g., Figure 3($c$) of the revised manuscript is now run on a vertical grid with 500 gridpoints (instead of the 200 gridpoints used previously). As Figure 3($c$) in the revised manuscript shows that time integrations run at higher resolution give the same result, we have applied a 5-window smoothing to the centroid locations in subsequent figures.

- In the limit of $W$ small, but integrated a long time, do the two PV anomalies rise together, or do they separate as seen here?

  The key point for this imposed ascent calculation is that, above a minimum value of heating (and hence $W$), if the initial anomalies have depth $D$, say, then after time greater than $D/W$ the anomalies will have separated. The steadily translating solution exists for all $W$ above this minimum value: the question is simply how long it takes to develop. In light of your comment, we have now added a discussion about the minimum heating required for the 'self-lofting' scaling estimates to be valid in §3.2 beginning on L403.

- Might help to label the color bar with units.

  Done: the colorbars in all figures in the manuscript now have units.

Figure 3: (Note this is now Figure 4 in the revised manuscript.)

- Is the radial location of the minimum temperature always at $r = 0$?

  Yes, as the minimum QGPV is centred on $r = 0$ and the temperature is calculated from the QGPV streamfunction, $T = (H_0 f/R)\partial_z \psi$. We have now added this in parenthesis on L347-348.

- Is this the absolute temperature minus the background temperature?

  $T$ here is the QGPV temperature anomaly as calculated from the QGPV streamfunction, $T = (H_0 f/R)\partial_z \psi$. The background vertically-varying temperature is $T_B(z)$.

- Again, hard to see the grey line.

  Done: the tracer centroid is now plotted in a solid black line and the anticyclone centroid in a solid blue line in all figures for improved readability.

- May want to label the units either near the color bar or in the caption.

  Done: the colorbars in all figures in the manuscript now have units.

- Text refers to Figure 3a on line 305, but there is no 3a in the actual figure. May want to label all the panels in this figure and other figures to help the reader.

  Done: all panels in all figures have now been labelled (a),(b), etc.

- On the right column, the labels say "Steadily translating solution v numerical solution". But this is $q$, not $v$, right?

  That is correct, the steadily translating solution is for the QGPV. The "v" in the titles of the right hand column of this figure was short for "versus" in the original manuscript. We have now spelt out the word "versus" in each of the subplot titles of Figure 4.

- For the right column, are the units consistent with the labeled values on the contours?

  The units for the QGPV solutions (steadily translating and numerical) in the right hand column of Figure 4 is $s^{-1}$. Previously, the contour interval for the colour-filled shading and the black contours was different: the contour intervals have now been made the same.

Line 333-4: I'm confused how the estimate for the PV anomaly is $f$. Why doesn't this depend on the heating rate, since in the limit of zero heating wouldn't there be no PV anomaly. I see you discuss this further in the last section, but it still doesn't make intuitive sense.

The reason why the PV anomaly is independent of heating is as follows: the PV forcing is proportional to $Q_s/D$ where $Q_s$ is heating and $D$ is depth and the time that this is applied at a given level is proportional to $D/W$ (hence $D/Q_s$), with the result that the PV anomaly is independent of $Q_s$ (see L366-3604). As the heating tends to zero, the PV forcing is weak but is applied at a given height over a long time. Therefore the vortex takes longer to reach $O(f)$ peak vorticity (see the related discussion on L825-826). As mentioned previously in this response, there is a minimum value of heating needed for our 'self-lofting' scaling estimates to be valid: we have now added a discussion on this in §3.2 beginning on L403.

Line 384: The diffusivities are constant, yet horizontal diffusivity in the stratosphere has been shown to vary, e.g., having minima associated with significant barriers to horizontal mixing associated with vortices (e.g., Haynes and Shuckburgh, 2000). Can the authors comment on whether you would expect similar diffusivity structures in these newly discovered tracer-driven vortices? Is there a way you could simulate this effect in your model by having diffusivity depend on model variables, such as PV gradient?

We take this opportunity to clarify that we included diffusivity solely as a numerical device to prevent sharp gradients in tracer from blowing up the code, rather than to represent any particular physical mixing process.

We agree that strong PV gradients surrounding vortices on scales of a few 100km or less (relevant here) can act as transport barriers, similar to the strong PV gradients at the edge of the polar vortex (cf. Haynes & Shuckburgh (2000) which focused on such large-scale features). That said, our axisymmetric framework essentially assumes isolation. Representing vortex stripping effects by a PV-gradient-dependent-$\kappa_h$ is in principle possible, however, it is unclear if this is a significant step towards reality as, in the real atmosphere with external shear/strain, the vortex in regions of weak vorticity would form elongated filaments that would eventually get stripped away, leaving a smaller vortex that persists. (See for example January 23's panel in Figure 5($a$) in Khaykin et al. (2020) which shows such a filament that got stripped away by January 31. Note that this filament is at a particular level [that is lower down in the vortex structure], and does not correspond simply to what would result from deformation by the quasi-horizontal flow.) Through the action of the external large-scale flow, vortices are likely to be substantially deformed and will therefore be substantially non-axisymmetric (see for example Figure 1($c$) or Figure 2 in Khaykin et al. (2022) showing Raikoke aerosol or, subject to reanalysis uncertainty, the PV structures in Figure 3 of Lestrelin et al. (2021)). Therefore, a promising avenue for our future research would be a suitably idealised three-dimensional calculation where non-axisymmetry can be included, so that effects such as transport barriers and vortex stripping can be studied.

Figure 4: (Note this is now Figure 5 in the revised manuscript.)

- May want to include labels for (a), (b), . . . , (f) on the figure

  Done: all panels in all figures have now been labelled (a), (b), etc.

- Are the units for heating correct, about $10^{-5}$ K/day or should they be K/s?

  Thank you for spotting the typo: the units have been corrected to K s$^{-1}$ in the text and figures.

- Hard to see tracer maximum and QGPV minimum symbols.

  We agree that the symbols were hard to see, and have now removed these symbols from all relevant figures of the manuscript as the region of tracer maximum and QGPV minimum is clearly visible by eye.

- The tracer distributions have what appears to be a very narrow maximum and then a broader elevated region, which is different from the observations. In the top and bottom cases there are actually two distinct maxima. Do you have any comment on this structure?

  The maximum at the top of the tracer structure is actually consistent with observations. See for example Figure 5($a$) of Khaykin et al. (2020): the snapshots of the tracer bubble shows that the highest attenuated scattering ratio profiles from CALIOP (e.g. the red/purple regions on January 16) are at the top of the tracer bubble. Physically, this is because $\partial_z w < 0$ at the upper boundary of the tracer filled region. The vertical velocity is strongest where the heating effect (i.e., tracer concentrations) maximises. Therefore, regions of high aerosol abundances would loft the most quickly to the top of the aerosol bubble.

  The structure of the tracer profiles in Figure 5 of our revised manuscript (as referred to in this comment, i.e., with a maximum at the top and then a broad elevated region) can be explained by the aspect ratio influence on vertical velocity. We explain this in the paper (L446-457), and re-state it here for clarity:

  – For a deep initial condition, $w$ is broader and shallower than the tracer. Most of the tracer is transported upwards.

  – For a shallow initial condition, $w$ is narrower and taller than tracer. The central region of tracer is transported upwards (and faster than the outer region).

  Thank you for pointing out the two distinct maxima in Figure 5($ae$). This is a numerical artifact that disappears when vertical diffusion is increased. In all figures in the manuscript, we have now increased the vertical diffusion by a factor of two: now, there is one broad maxima at the top of the tracer structure.

- It appears the temperature minimum is co-located with the QGPV minimum. Is that correct? If so, this is also different from observations.

  In our model, the QGPV minimum is co-located with the temperature minimum because the minimum QGPV occurs at the top of the anticyclone where the strongest heating rate (i.e. tracer abundances) are. This is different to the pattern in composites from ERA5 PV, subject to reanalysis uncertainty. However, the anticyclone centroid and tracer bubble centroid line up with the location where $T = 0$ in the upper dipole temperature anomaly (i.e., between the cooling above and warming below), which is in agreement with reanalysis composites of temperature and PV.

  Previously when the vertical diffusion was smaller, the QGPV minimum and temperature minimum were picking out the small-scale, narrow features at the top of the tracer structure (see our response to your previous comment). Now that we have increased the vertical diffusion by $2\times$, there are broad regions with negative temperature in the cooling anomaly and with negative values of QGPV in the anticyclone.

  We have now made the following two changes: (1) To make clear where the temperature minimum is, we have now included the temperature contours in Figure 5($ace$). (2) To focus on the overall tracer-vortex structure, we have replaced the line marking the vertical location of the tracer maximum with a line marking the vertical location of the tracer *centroid* in all figures showing vertical slices of QGPV. In such figures, we have removed the line marking the vertical location of QGPV minimum,

[Figure]

Figure R1: $(u, v, w)$ profiles integrated from the standard initial condition after 50 days without thermal damping (i.e., $\alpha = 0 \, \mathrm{day}^{-1}$) and without vortex stripping. These are the velocity profiles for the configuration shown in Figure 5($ab$) in our manuscript.

> as the broad takeaway from these figures is that the tracer centroid is inside the anticyclone. In figure panels showing vertical slices of the tracer, we have removed the line marking the vertical location of tracer maximum and now only show the vertical location of the anticyclone *centroid*, again to make the general point that the anticyclone centroid lies inside the tracer cloud.

It might be useful to plot a schematic diagram, similar to Figure 1b, for this simulation. This would help the reader to compare more directly with observations.

Done. We have now added a new schematic with accompanying text summarising the key behaviour our model describes in Figure 9 at the end of §3.5. Panel ($a$) shows the initial configuration, with panels ($bcd$) showing key model behaviour: ($b$) without vortex stripping and without thermal damping, ($c$) with vortex stripping and without thermal damping, ($d$) with vortex stripping and with thermal damping.

Is there a significant radial wind in these simulations? It would be nice to see the complete $u, v, w$ structure.

In the axisymmetric balanced dynamics being considered the radial velocity $u$ is determined from $w$ by mass continuity (and $w$ is determined by (7)). Maximum values of $u$ are typically of order $0.1 \mathrm{ms}^{-1}$ and in most of the domain much less. For these reasons we have not included a plot of $u$ in the revised manuscript, but we do now quote typical values on L435. However we have included a plot in this response for information (Figure R1). Whilst the $u$ field plays an important role in the axisymmetric calculation since it is the $(u, w)$ field that transports the tracer, note that any $u$ field predicted by an axisymmetric calculation will be strongly perturbed by any three-dimensional asymmetry. It is these three-dimensional effects that are likely to determine the $u$-field in any observed case or realistic model (unless axisymmetry is being enforced).

The vortex stripping experiments are used to help remove tracer that occurs well below the maximum. How much of a role does vertical diffusion play in this remnant low-level tracer? What happens if you run the model with very small or very large vertical diffusion?

Vertical diffusion plays no role in our vortex-stripping adjustment, which is motivated by vortex stripping experiments (e.g., Marriotti et al., 1994). In §3.4, the tracer in regions where $q < q_{\mathrm{crit}}$ is simply removed.

Figure 5: (Note this is now Figure 7 in the revised manuscript.)

[Figure]

Figure R2: Wind profiles $(u, v, w)$ after 50 days of time integration from the standard initial condition, with thermal damping $\alpha = 0.1$ day$^{-1}$ and with vortex stripping adjustment applied. Here, $\rho_0$ is held constant (i.e., these wind profiles correspond to Figure 8($acd$) of our manuscript).

- May want to include labels for (a), (b), (c), (d) on the figure

  Done: all panels in all figures have now been labelled (a), (b), etc.

- On line 442 I think you have the panels reverse: (a) should be without, (b) should be with

  Done: this typo has now been corrected on L501.

- Does the temperature dipole structure have better agreement to observations in this case than in the previous? Is there cooling above and warming below the QGPV minimum?

  See above for our response to your previous question about the temperature dipole in Figure 5 of revised manuscript: for all subsequent figures, we have now plotted the vertical location of the anti-cyclone *centroid* in panels showing tracer profiles over time and tracer *centroid* in panels showing PV profiles over time (i.e., we no longer plot the QGPV minimum and tracer maximum). As the vortex stripping adjustment makes the PV anomaly more vertically compact, the agreement with observations is improved: the anticyclone centroid falls within the $T = 0$ region between the upper cold anomaly and lower warm anomaly (not shown in the manuscript, but checked separately).

- Again, a schematic like Figure 1b would be nice.

  Done. As mentioned previously, we have now added a schematic in Figure 9 and accompanying summary of key behaviour in our model in at the end of §3.5.

Figure 6: (Note this is now Figure 8 in the revised manuscript.)

- Again, may want labels on the individual figure panels.

  Done. All panels in all figures have now been labelled (a), (b), etc.

- Again, it would be interesting to see the structures in the wind fields for this case.

  We have provided the wind profiles after 50 days of time integration from the standard initial condition, with thermal damping $\alpha = 0.1$ day$^{-1}$ and with vortex stripping adjustment applied in Figure R2. $\rho_0$ is held constant, corresponding to Figure 8($acd$) of our manuscript. Please see our reply above to your comment about wind fields in Figure 5 for a discussion of the role of $u$ in our model.

- Again, the temperature dipole does not agree with Figure 1b. The minimum peaks at the same location as the PV minimum. Any idea why this is the case?

  See above for our response to your similar comment about Figures 5 and 7 of our manuscript. (Note that for improved readability of Figure 8, we have removed the temperature contours from panels ($df$).)

The discussion of the instability in the $x$-$z$ plane was very interesting. I wonder if it would be beneficial (not for this paper, but for future work) to also perform the 2-D analysis in the $x$-$y$ plane (e.g., shallow water system) where a heating tracer is initially placed in an irrotational flow. The heating could increase the layer thickness and cause rotation via geostrophic balance, which would help to contain the tracer against horizontal shearing flow. This wouldn't model the lofting effect but could help to understand some of the self-organizing flow.

Thank you for this suggestion. We will certainly bear it in mind for future studies. As is implicit in your suggestion, the challenge in studying this system is that it is fundamentally three-dimensional: the 2D (horizontal × vertical) approach here misses the 'vortex-dynamics' aspects of the evolution, a 2D (horizontal × horizontal) approach would miss the lofting which is fundamental to the structure of the vortices. Ultimately, a suitably idealised 3D model would probably be the most promising avenue for our future research.

Line 673: There were at least three vortices from the 2019/2020 Australian wildfires. I think you mean the largest vortex, since the others ascended at a slower rate and didn't go as high or last as long.

Yes, we meant the main vortex and have now edited the sentence on L772-775.

Lines 687-689: Might be useful to quote your typical azimuthal velocity and temperatures produced in your simulations and compare them to observations.

Done: we have now added a sentence on L788-789 saying that typical $v$ and $T$ values from our time integrations and from our scalings are consistent with composites in Khaykin et al. (2020).

Lines 720-722: Some discussion of vortex resistance to shear was also presented in Allen et al. (2020), https://doi.org/10.1175/JAS-D-20-0131.1 which was a follow-up to Kablick et al. (2020) and provided more detailed analysis of dynamical properties of the main anticyclone associated with the Australian New Year event, in addition to examining one of the 2017 Pacific Northwest Event plumes.

Thank you for bringing this paper to our attention. We have added a sentence on L861-864 about Allen et al. (2020)'s discussion of the behaviour of anticyclonic vortices in external shear. We have also cited Allen et al. (2020) in the introductory paragraph of the paper on L33 given that this paper shows preliminary evidence for a vortex originating from the 2017 Canadian wildfires.

**Technical Corrections**

Line 3: should be "carbon monoxide"

Done: this now reads "carbon monoxide".

Line 30: should be "carbon monoxide"

Done: this now reads "carbon monoxide".

Line 31: Should be "have shown" to match the subject "fields"

Done: this now reads "have shown".

Line 148: "a r a" sounds funny. Maybe remove first "a"

Done: this now reads "$r$ a".

Figure 1 caption: "and volcanic eruptions" can be removed before the first set of references

Done: we have removed "and volcanic eruptions".

Line 330: remove "is" from "ascent is tends"

Done: we have removed "is".

**Response to Bernard Legras (Reviewer 2)**

Review of "How heating tracers drive self-lofting long-lived stratospheric anticyclones: simple dynamical models," by Kasturi S. Shah and Peter H. Haynes.

This is an interesting work addressing a new type of long-lived coherent vortices recently discovered in the stratosphere following the injection of large amount of smoke by pyro-convection resulting from wildfires.

It happens that these vortices are maintained by a localized heating source produced by the absorption of incoming solar radiation and are dissipated by the longwave radiative relaxation of the accompanying thermal anomaly. The study of response to localized heating in the atmosphere is not new in the context of convection and tropical cyclones but here the context is of a resulting stable ascending vortex which was not considered before.

This study is based on the reasonable approach of the maximum simplification. It considers an axisymmetric system where heating is proportional to the mixing ratio of an idealized tracer and a Newtonian relaxation modelling the longwave radiative damping of the temperature anomaly.

Thank you for your thorough review of our manuscript. We have provided a point-by-point response to your review below. All line numbers in this response file refer to the *revised* manuscript, unless otherwise indicated. We have also uploaded our revised manuscript as well as a marked-up version of the revised manuscript in which additions to the text are marked in deep green colour.

A very similar approach was adopted by another study, Podglajen et al. (2023) (hereafter P2023), currently under review for QJRMS and of which I am a co-author. . Most of my comments will be based on a comparison with this study and the previous works of Lestrelin et al. (2021) (hereafter L2021) and Khaykin et al. (2020) (hereafter K2020).

Since receiving your review we have now read P2023 and considered its relation to our own work. Certainly, the combined use of 3-D numerical simulations and the elegant Burger's-like model overcomes some of the shortcomings of our own approach of a balanced axisymmetric model, for which we had resort to a heuristic and formally inconsistent quasigeostrophic assumption to make progress beyond the initial evolution. The Burger's analogy provides a convincing explanation of why a balanced model approach to the heating tracer problem breaks down without the introduction of some extra physics, such as vertical diffusion of tracer. The insight from P2023 notwithstanding, we think there are still plenty of other questions that merit future research, including the strength of radiative effects, nature of environmental flow, influences on the viability/lifetime of the vortices, the physics determining the interior PV structure of the vortices, etc. We believe there is some virtue in both P2023 and our study being published: their focus is not identical and they should both stimulate further research on this topic's many outstanding questions. The approach we have taken in revision is to incorporate various references to and comments on P2023, where appropriate, and to clearly acknowledge where P2023 has influenced our own thinking in revising the paper. Specifically we have made the following changes:

1. We have added (i) a new subsection §3.6 which provides a perspective on P2023 and our study, highlighting similarities and differences, and (ii) a discussion of the initial condition in P2023 and our study in §5 "Discussion and Conclusions" beginning on L832. Our goal with these changes was to recommend that readers read P2023, which in some respects goes well beyond our own work.

2. While we were clear about the limitations of the QG approach in the first version of our manuscript (see §3.2, §3.6, §5 of the original manuscript), in light of your comments we have added further discussion about it in this revised manuscript: for instance, in §3.6, we have also added a discussion of the

restriction of our study to the QG framework (including your point below about the neglect of vertical advection of PV).

3. With regard to the effect of thermal damping, in the previous version of the manuscript, we discussed the effect of thermal damping primarily when it accompanied vortex stripping. In light of your comments on the effects of thermal damping and the results of P2023 on this we have re-considered the effects in our own model. We have noted that increased ascent of the leading tracer edge as a result of thermal damping has been identified by P2023 and have explained how this fits with our own revised scaling arguments and re-examined model results. We have now updated the conclusions in light of these findings and clarifications.

**General comment**

The manuscript seems to take as granted that the ERA5 reanalysis provide "observations" of the ascending vortices and in particular of the potential vorticity (PV) distribution. It is indeed remarkable that the smoke clouds as detected by the lidar CALIOP correspond very closely to the anomalies of PV in ERA5 (K2020 & L2021), even when two vortices are crossing at different altitudes (P2023). However, the ERA5 does not assimilate aerosol data and therefore the anomalous heating is absent. It is only informed by the assimilation of IASI infrared radiances and GPS radio-occultations which constrain the temperature (K2020). In principle the exact knowledge of the temperature distribution implies the knowledge of the equilibrated wind field but assimilation only provide a limited knowledge and vey different PV distributions can produce a fairly similar temperature distributions.

We agree that our paper did not emphasise enough the uncertainty regarding the details of the PV structure associated with the ascending vortices. Specifically, that while observational information on the temperature field has fed into the ERA5 assimilation process, this provides a weak constraint on the PV and more generally that statements about reanalysis PV should make clear the PV field is subject to uncertainty. In response to your comment, we have now added text about the reanalysis uncertainty in §1 on L95-105, as well as in the main body of the paper where relevant to the discussion of results.

Introducing an active tracer in a manner similar to that used in the present study, P2023 show that the PV distribution produced by fully resolved 3D simulations of the primitive equations is very different from that "observed" by ERA5. The main reason is that the tracer tends to form a front at its top, followed by a tail (a pattern visible on figures 4 and 5 of the manuscript) and that this has a determining impact on the PV field generated and maintained by the heating. This important point has been, in my view, totally missed in the manuscript , and deserves some further examination by the authors, in particular regarding the scaling laws. P2023 also argue that the heating tracer dynamics is associated with a zero-PV, an aspect which is misrepresented here in part due to the restriction to the quasigeostrophic framework.

We are interpreting the 'tracer tail' referred to in this comment as the deepening of the tracer structure over time. 'Tracer tail' structures, arising because of the difference in vertical velocity across the tracer cloud, were indeed evident in our results in Figures 4 and 5 and we commented on them (e.g., at the start of §3.4 of original manuscript). The reason we chose not to focus on the impact the trailing tracer structures have on the PV field is two-fold.

- First, our schematic model presented in Section 3.1 is based on the *observed* structure of the tracer field. In aerosol observations, the tracer moves upwards in a localised bubble. There is little evidence for a deepening cloud. In observations (e.g. Khaykin et al 2020, Figure 5) the tracer seems to move upward in a confined bubble. There is some deepening of the bubble, e.g. from 16 Jan to 31 Jan, but that seems quite different from the changing structure of the tracer cloud seen in P2023 Figure 8 or in our Figure 5 (revised version) where the base seems almost stationary and the depth of the entire cloud increases over the entire evolution.

On L444-445 we are now explicit that we do not take account of the tracer tails in the scaling arguments, justified by them not being observed. We have now been clear about this disagreement between observations and model in the revised manuscript, and that it needs to be resolved.

- Second, we anticipate that these trailing features would be dispersed and diluted by the large-scale flow via 3D effects that our axisymmetric model does not capture (see the start of §3.4). We see 'vortex stripping' (in its generalised sense to include possible effects of vertical shear as well as horizontal deformation) as a key mechanism that is likely to affect the peripheral structure of the vortex and which is neglected in the axisymmetric model. Therefore, we chose to incorporate a vortex-stripping adjustment in §3.4.

In response to Reviewer 1's request for a schematic, we have added Figure 9 at the end of §3.5 where we now discuss the role of the trailing tracer structures in conjunction with this newly added schematic. Additionally, we have now noted on L602-605 that P2023 also find trailing structures in tracer.

Regarding the last sentence of this comment about zero-PV, we have made it clear in §3.6 in our revised manuscript that P2023 have made a valid prediction of near-zero PV (L580-583). Our QG theory makes some useful predictions of balanced dynamics when PV is small but non-zero, e.g., that our $q \sim O(f)$ scaling suggests that PV will be small for an anticyclone. To aid reader interpretation of the results of both studies, we have now added a discussion of the correspondence between QGPV of $O(f)$ and near-zero full PV following Berrisford et al. (1993) (L583-586). We have also noted that P2023's numerical simulations show only small regions of zero or negative PV, with the core of the anticyclone in 3D simulations apparently having 'small but non-zero' PV (L600-602).

**Specific comments**

L150: If one uses a log-pressure coordinate, the height scale in the continuity equation should not be different from that of the hydrostatic equation. This is presented as a "non-standard" feature but I would like to see some justification. Pretending to decouple the variation of density and the stratification looks a bit weird in my view.

This different scale height is simply a convenient device to allow us to combine Boussinesq ($H \to \infty$) and non-Boussinesq ($H = H_0$) in the same set of equations.

The prediction shown in Fig1a is first discussed as contradicting the observed pattern in Fig1b. There is obviously no contradiction if the pattern is a vertically moving vortex as indicated on L268. I find this presentation more confusing than illuminating. This is also discussed in L2021.

We have now removed any mention of the words "inconsistency" or "contradiction" from the manuscript. We have also made a number of edits to clarify that the "observed" PV structure is different to the theoretical prediction for *stationary* heating, by way of edits to the text in §3.2, edits to the title of panel (a) in Figure 1, and edits to Figure 1's caption.

We designed Figure 1 in light of comments made in the literature. For instance, K2020 notes the discrepancy in the PV field structure and the expected response to stationary heating: "More detailed studies will be necessary to understand fully the accompanying dynamical processes, in particular how a single sign vorticity structure emerges as a response to heating." In response to Reviewer 1's request to include additional schematics describing the physical mechanisms in §3.3 and §3.4, we have chosen to keep the schematic in Figure 1 and to add an additional schematic as Figure 9 at the end of §3.5.

The ICs chosen in 3.2 are a tracer blob without initial PV anomaly. P2023 argue that the mass uplift due to pyro-convection means PV dilution at the top and therefore an initial PV anomaly accompanying the tracer

anomaly, which is ignored here. The importance of the initial PV anomaly for the subsequent development is discussed in P2023.

We agree that the injection of tropospheric air into the stratosphere will by itself give an anticyclonic PV anomaly and have cited P2023 on this point. However, in our view, whether a low PV initial condition is the only 'relevant' initial condition is open to question. We have now laid out this argument in §5 on L832-847.

a,b,c labels of panels are missing on Figure 2 and other figures. The fonts on the small panels (axis, colorbar, legends) are too small and should be increased for readability. Take into account that the figures will be smaller in the final publication than in the manuscript.

All panels of all figures are now labelled (a), (b), etc. The fontsize in all figures has now been increased.

L330: The rise of a shallow region of tracer depends on the distribution of tracer as a function of radius. A disk of uniform tracer will rise uniformly as least in the initial stage. The ascent depends on the heating which depends on the tracer.

The rise, i.e. the rate of ascent, looks different in different coordinate systems. Of course, if $\dot{\theta}$ is imposed as horizontally uniform in $\theta$-coordinates then the rate of ascent measured in $\theta$-coordinates will be horizontally uniform. However, the rate of ascent in $z$-coordinates will not: the relation between $z$ and $\theta$ will change as the system responds to the heating. This would be more apparent if P2023's 2D model were plotted in geometric coordinates.

The numerical simulations and the theoretical discussion in P2023 show that the PV decreases exponentially with time to zero in the frontal region with a time scale that does not exceed a few days for realistic parameters, so that the validity of a QG model might be actually very short. It should be noted that the "observations" of PV from ERA5 also show zero and even slightly positive PV at the center of the main vortex of the 2020 event (L2021).

Thank you for pointing out the exponential approach to zero PV in P2023. We hypothesise that the diffusion we add to our tracer equation is not large enough to halt the contraction in scale before the grid scale is reached, which is why we see a linear in time approach rather than exponential.

We don't dispute that the quantitative validity of the QGPV model will apply only for a relatively short time. In fact, we had already noted the shortcomings of the QG approximation in various places in the original manuscript (see §3.2, §3.6, §5 of the original manuscript). We have now added a discussion about limitations of the QG framework in §3.6 (e.g., L587-595).

We have now cited the near-zero PV composites from L2021 on L780.

Actually, the tracer problem with no temperature relaxation and neglecting lateral diffusion can be solved exactly as a Burgers' equation in the transformed momentum coordinate system as shown in P2023. Then the PV can also be obtained as the corresponding hyperbolic equation can be solved by characteristic method.

Having now read P2023, we have included a description of the method of solution in §3.6.

L356-363: Another effect, which cannot be seen here, is to accelerate the ascent. The main reason is that the radiative damping reduces the decay of the tracer. See P2023.

The accelerated ascent of the leading edge of the tracer structure *is* produced by our model. The effect is most evident when thermal relaxation is included *without* vortex stripping adjustment (as the vortex stripping adjustment removes low-lying tracer in regions of weak anticyclonic PV). In the original manuscript, we had chosen to show results with thermal damping *and* vortex adjustment to streamline the storyline (as one of the arguments we make in our study is that the trailing tracer features are unlikely to survive in the real atmosphere where 3D effects such as dispersion by the large-scale flow, vortex stripping, etc., are at play). In light of your comment and having now read P2023, we have added a discussion of results with thermal

damping *without* vortex adjustment at the end of §3.3 and in Figure 6 in our revised manuscript. There, we explain that the finding that thermal damping leads to increased ascent of the leading edge of the tracer structure originated with P2023, and comment on the qualitative agreement between P2023's findings and ours. Given this addition, we have also restructured the thermal damping arguments in §3.2 and removed some sentences for clarity.

L393-404: A main limitation of the QG model is that the PV is not advected by the vertical velocity. There is implicitly advection of the temperature but not of the vorticity, so one may expect that the PV anomaly will lag behind its true location and be deformed.It is likely that all the discussion contained in this paragraph is strongly affected by this limitation.

We first note that the paragraph to which this comment refers (on L393-L404 of the original manuscript) discusses the structure of the vertical velocity based on the initial condition of the tracer using scaling arguments arising from equation (7) in our manuscript. The possibility of the PV anomaly lagging behind its true location would not affect these arguments, as (in our model) PV does not appear in equation (7). Of course the subsequent evolution of the vertical velocity, and hence the evolution of the tracer, the heating and the PV is affected by the assumption of quasigeostrophy. To gain more information about the implication of the quasigeostrophic assumption we formulated a quasigeostrophic version of P2023's (17a). On integrating two sets of equations – (A) our modified version of P2023's (17a) and P2023's (17b) and (B) P2023's (17ab) – we find that the diabatic forcing term plays a major role in reducing the PV in the leading anticyclonic edge while the vertical advection term plays a role in deepening the cyclonic PV in the trailing edge. Accordingly, we have now added a discussion of the role of neglect of vertical advection in §3.6 on L587-595.

L415: Although this question was investigated, the cyclonic PV has not been seen so far in the tail of the vortices in the ERA5. However, it should be noted again that PV is forced in the model by the assimilation of temperature information and that this process does not need to satisfy the integral properties of PV conservation. It is also possible that the cyclonic PV is continuously washed out and dispersed by the vertical shear and cannot be detected.

We have now added discussion about how PV is forced in ERA5 in §1 and about how the lack of evidence of cyclonic PV in reanalysis is subject to uncertainty on e.g., L479-482. (We mentioned the possibility of cyclonic PV being dispersed by the large-scale flow on L425-431 of the original manuscript to motivate our vortex-stripping adjustment.)

It is interesting to note that Doglioni et al. (2022)'s numerical simulations show a lower-level positive PV anomaly as well as a negative PV anomaly co-located with the aerosol (see L480-481). These results are from a model, which has the advantage that they are not subject to the same uncertainty as reanalysis, though of course they are subject to other uncertainty, including whether the apparent low-level cyclone would or would not be observed at other times during the evolution of this vortex or for other vortices.

Section 3.4: What is seen in this section is essentially what has been implemented in the model and so the results, which display a cropped version of the patterns seen in previous section, are not really surprising. It should also be noted that there are two types of stripping involved her. One is the horizontal stripping studied by Mariotti et al., 1994 and the other one is the vertical stripping due to the vertical shear. In the case of the main vortex of the 2020 event, the two effects played a role at different stages of the evolution.

In response to the first sentence of this comment, it not so obvious that an upward propagating anticyclone would be maintained over a long period with the vortex stripping adjustment. We take this opportunity to clarify that our main motivation for including Section 3.4 is that the trailing tracer features (and the weak PV they generate) are, in our view, unlikely to influence the evolution of aerosol-filled vortices in the real atmosphere because they are stripped away by the 3D background flow under the influence of, for example, horizontal and vertical shear. An interesting avenue for future research would be a generalised Marriotti et al (1994) calculation in which a QG vortex is subject to horizontal and vertical shear.

In response to your point about vertical stripping, we have clarified that both horizontal and vertical shear

could be at work in the aerosol-filled vortex context in §1 on L61 and L72-76 and have added text in §3.4 on L489-490 making clear that both horizontal and vertical stripping are at work. We have also added a sentence on L861-864 in §5 that references K2020, L2021 and Allen et al., 2020, as all three mention the behaviour of smoke-filled vortices in vertical shear flows.

Section 3.6: The failure of the numerical solution of the inversion problem is not totally surprising. Although in principle the PV never reaches exactly zero, it can be so close that the numerical procedure breaks down. It is a bit surprising that the trend to zero is linear meaning a catastrophe at finite time as it is expected to be exponential.

See above our note about the role of diffusion in potentially arresting the exponential approach to zero PV. (Note that our revision of the manuscript has reorganised the original §3.6 on the non-QG Eliassen problem to be §2.1 now.)

Section 4: This is an interesting development which seems to bear some similarity with the radiative instability at the top of a cirrus cloud as studied, e.g., by Dinh et al., 2010. I am however wondering of the relevance to the present problem as the scale of the instability is not at all that of the observed vortices which is in the mesoscale range, 500 to 1000 km. This approach neglects the role of the initial injection of mass as stated above.

We have added a mention of Dinh et al. (2010) on L708-711.

The motivation for §4 is more general than smoke-filled vortices: it considers a fundamental configuration of a homogenous layer of tracer. To clarify the general motivation of this section, we have deleted the sentence "When the initial condition for the tracer distribution was thin in the vertical relative to the horizontal, it was shown that a plume of tracer, narrower in the horizontal than the original distribution, rises upwards from its centre (Figure 4(e))." and edited text at the start of §4.

With respect to the scale of the vortices compared to the scale of the tracer structures in our Figure 10, the plumes would be wider if the initial horizontal layer were thicker (which we now mention on L724-725). Moreover, the analysis presented here really examines the *initial* break-up of the homogenous layer. It is certainly possible that the tendrils of tracer in a 3D flow would eventually pinch off and coalesce to form tracer bubbles of $O(500 - 1000)$ kms across. On L727-728, we have now added such 3D effects that the 2D stability analysis does not capture.

L694: This is just not true when all effects are accounted as explained above.

Having now read P2023 and done some additional thermal damping analysis, we have now rewritten the sentence to which this comment refers ("This lack of influence of thermal damping on the dynamics is supported by scaling arguments that find that the length of time these vortices spend at a given level is less than the timescale of radiative damping in the lower stratosphere.") and its neighbouring sentences to reflect our latest findings. We have also made a number of edits to the thermal damping-related text in the manuscript, including clarifying in §3.2 that the scaling arguments are derived from the simplified configuration of an upward moving heating and, as stated previously in this response, adding results to the end of §3.3 showing the role of thermal damping in the fully coupled model *without* vortex stripping adjustment.

We note that P2023 appear to mention use of a thermal damping rate of 2.2 day$^{-1}$ in Appendix A2 which seems very large and we wonder if this is a typo.

There are many other points to be reworked in the conclusion in the lights of the comments made here.

We have now made a number of edits to the 'Discussion and Conclusions' (§5) for clarity. We have also added the following: a discussion of different viewpoints on the initial condition used in P2023 and our study, clarification of the findings about thermal damping, a sentence about correspondence between QGPV and full PV, a sentence about the role of vertical shear.

**Minor comments**

l.30: The co-location with other species is also shown clearly in K2020.

Done: K2020 has been cited here.

L45: It should be referred to Manney et al., 2006 for "frozen in" anticyclones

Done: Manney et al., (2006) has been cited here.

L159: Tb should be TB

Done: $T_b$ has been corrected to $T_B$.

L172: Perhaps quote here the condition that PV should be non zero and of the sign of f to solve the Sawyer-Elliassen equation

Done: we have mentioned this condition on L188.

L195: Add "anomaly" to temperature.

Done: the word "anomaly" has been added here.

L235: Quote also Davies, 2015

Done: Davies (2015) has been cited here.

L351-354: This is repeating L317-319

Done: we have removed the repeated information.

L525: Remove one the

Done: repeated word "the" has been removed.

L545: s should be h in the formula

Done: $\hat{s}$ has been corrected to $\hat{h}$.

References

Davies, H. C.: The Quasigeostrophic Omega Equation: Reappraisal, Refinements, and Relevance, Monthly Weather Review, 143, 3–25, https://doi.org/10.1175/MWR-D-14-00098.1, 2015.

Dinh, T. P., Durran, D. R., and Ackerman, T. P.: Maintenance of tropical tropopause layer cirrus, Journal of Geophysical Research, 115, D021014, https://doi.org/10.1029/2009JD012735, 2010.

Khaykin, S., Legras, B., Bucci, S., Sellitto, P., Isaksen, L., Tencé, F., Bekki, S., Bourassa, A., Rieger, L., Zawada, D., Jumelet, J., and Godin-Beekmann, S.: The 2019/20 Australian wildfires generated a persistent smoke-charged vortex rising up to 35 km altitude, Commun Earth Environ, 1, 22, https://doi.org/10.1038/s43247-020-00022-5, 2020.

Lestrelin, H., Legras, B., Podglajen, A., and Salihoglu, M.: Smoke-charged vortices in the stratosphere generated by wildfires and their behaviour in both hemispheres: comparing Australia 2020 to Canada 2017, Atmos. Chem. Phys., 21, 7113–7134, https://doi.org/10.5194/acp-21-7113-2021, 2021.

Mariotti, A., Legras, B., and Dritschel, D. G.: Vortex stripping and the erosion of coherent structures in

two-dimensional flows, Physics of Fluids, 6, 3954–3962, https://doi.org/10.1063/1.868385, 1994.

Podglajen, A., Legras, B., Lapeyre, G., Plougonven, R., Zeitlin, V., Brémaud, V., and Sellitto, P.: Dynamics of diabatically-forced anticyclonic plumes in the stratosphere, https://doi.org/10.22541/essoar.169603596.62706666/v1, 30 September 2023, revised, sub judice in Quart. J. Roy. Met. Soc.